# The normal human lymph node cell classification and landscape defined by high-dimensional spatial proteomics

Maddalena M. Bolognesi[1,2☯], Lorenzo Dall'Olio[3☯], Giulio Eugenio Mandelli[4,5], Luisa Lorenzi[4,5], Francesca M. Bosisio[6,7,8], Ann M. Haberman[9,10], Govind Bhagat[11¤], Simone Borghesi[12], Mario Faretta[13], Gastone Castellani[14☯], Giorgio Cattoretti[15☯*]

1 Istituto di Bioimmagini e Sistemi Biologici Complessi (IBSBC) – CNR Via F.lli Cervi, Segrate, Italy, 2 NBFC, National Biodiversity Future Center, Palermo, Italy, 3 Laboratorio di Data Science e Bioinformatics, IRCCS Istituto delle Scienze Neurologiche di Bologna – AUSL BO Ospedale Bellaria, Bologna, Italy, 4 Pathology Unit, Department of Molecular and Translational Medicine-DMMT, University of Brescia, Brescia, Italy, 5 Pathology Unit, ASST Spedali Civili Di Brescia, Brescia, Italy, 6 The Leuven Institute for Single-cell Omics (LISCO), KULeuven, Leuven, Belgium, 7 Translational Cell and Tissue Research Unit, Department of imaging and pathology, KULeuven, Leuven, Belgium, 8 Pathology Department, University Hospital of Leuven, Leuven, Belgium, 9 Department of Immunobiology, Yale University, New Haven, Connecticut, United States of America, 10 Department of Laboratory Medicine, Yale University, New Haven, Connecticut, United States of America, 11 Pathology, Columbia University Irving Medical Center and New York Presbyterian Hospital, NewYork, New York, United States of America, 12 Department of Mathematics and Applications, University of Milano Bicocca, Milan, Italy, 13 Department of Experimental Oncology, European Institute of Oncology IRCCS, Milan, Italy, 14 Department of Experimental, Diagnostic and Specialty Medicine, University of Bologna, Bologna, Italy, 15 Pathology, Department of Medicine and Surgery, Universitá di Milano-Bicocca, Monza, Michigan, Italy

☯ These authors contributed equally to this work.
¤ Current address: Pathology, Saint Louis University School of Medicine, Saint Louis, MO, USA
* giorgio.cattoretti@unimib.it

## Abstract

Lymph nodes (LN) are key secondary lymphoid organs (SLO) for a coordinated immune response. They have been extensively characterized by numerous investigative techniques chiefly as single cell suspensions because they are composed of vagile yet crowded hematolymphoid elements, unfriendly to spatial tissue organization-saving techniques. We comprehensively classify in situ all cells of 19 human LN free of pathology with a 78-marker antibody panel, an hyperplexed cyclic staining method, MILAN, and an analytical bioinformatic pipeline, BRAQUE. A total of 77 cell types were classified, encompassing T, B, innate immune and stromal cells. CD4 and CD8 T-cells were classified into 27 unique subsets by leveraging the expression profiles of TCF7, the presence of co-inhibitory receptors and the spatial distribution. CD5 and TCF7 expression defined novel B-cell types. CD27 + mature B-cells occupied previously unrecognized nodal spaces non-overlapping with the cortex and the plasma-cell rich medullary cords. Type 2 conventional dendritic cells were located in nodular paracortical aggregates. Statistically controlled pairwise neighborhood analysis showed sparse cell-cell interactions, known and new neighbors, established and novel LN landscape niches. A high-dimensional proteomic interrogation of the normal

---

**Data availability statement:** This publication is part of the Human Cell Atlas - www.humancellatlas.org/publications Primary data (images .tif files, .csv files) have been deposited at the Human cell Atlas data portal: https://data.humancellatlas.org/. They are available at: https://explore.data.humancellatlas.org/projects/b10cd314-3e71-4437-9a16-77028d243e81 Analysis data have been deposited at UNIMIB Digital Commons repository Bicocca Open Archive Research Data (BOARD): Bolognesi, Maddalena; Dall'Olio, Lorenzo; Borghesi, Simone; Castellani, Gastone; Cattoretti, Giorgio (2024), "The normal Lymph Node. BRAQUE analysis files & Raw Data", Bicocca Open Archive Research Data, V1, doi: 10.17632/3ntbp3zdzh.1 https://board.unimib.it/datasets/3ntbp3zdzh/1.

**Funding:** Regione Lombardia POR FESR 2014–2020, Call HUB Ricerca ed Innovazione: ImmunHUB. (GCat and MMB) EU Horizon 2020 programme (GenoMed4All project #101017549, HARMONY and HARMONY-PLUS project #116026) (GCast) The AIRC Foundation (Associazione Italiana per la Ricerca contro il Cancro; Milan, Italy; projects #26216. (GCast) KUL INTERNE FONDSEN MIDDEL-Zware infra-structuren EMH-D8191-AKUL/19/30 I005920N (FMB) FWO Fundamenteel Klinisch Mandaat EMH-D8972-FKM/20. (FMB) The National Recovery and Resilience Plan (NRRP), Mission 4 Component 2 Investment 1.4 - Call for tender No. 3138 of 16 December 2021, rectified by Decree n.3175 of 18 December 2021 of Italian Ministry of University and Research funded by the European Union – NextGenerationEU; Project code CN_00000033, Concession Decree No. 1034 of 17 June 2022 adopted by the Italian Ministry of University and Research, CUP B83C2200293000, Project title "National Biodiversity Future Center - NBFC". (MMB)" The funders had no role in study design, data collection and analysis, decision to publish, or preparation of the manuscript.

**Competing interests:** The authors have declared that no competing interests exist.

human LN provides spatial allocation of known cell types, novel interactions and the landscape organization.

## Introduction

The anatomical structure of the lymph node (LN) [1] is instrumental for its function, primarily to promote the interactions of cells with antigens or other cell types within discrete niches [2]. Recently, the cells composing the lymphoid tissue and the mutual interactions have been extensively investigated with high-capacity molecular and immunological methods at the single cell level. Such studies have produced a granular classification of the immune cells in various organs, including the LN, and along human immune system development [3], with some investigations specifically focusing on individual cell types [4]. The common goal of these initiatives is to build a Human Cell Atlas [5] to be used as a framework for a better knowledge of normal tissue architecture and cellular function in its tissue context and which could potentially help in the development of novel therapies for a variety of diseases.

Starting in the 80's, in situ immunostaining techniques have contributed to establish a static view of the immune architecture of the human LN; more recently, the use of various methods for the detection of multiple antigens at once [6] have provided a topographic view of the immune response, both for hematopoietic [7,8] and non-hematopoietic LN cells [9]. However, the cell composition of the LN has not been fully elucidated as a whole with in-situ methods.

We undertook a spatial single cell classification of a large number of LNs, representing variegated aspects of normalcy, with a panel of 78 antibodies and a novel image analytical pipeline, BRAQUE [10]. We here provide a granular purely phenotypic classification of established and novel cell types populating the human LN. We then established the relationships of each cell type with the others via a robust statistical approach and, lastly, we defined the regional landscape of the human lymph node based on the selective distribution of neighboring or distant cell types. In such a way we provide a reference database from which to extrapolate the effects of sustained immune challenges or neoplastic transformation.

## Results

To classify all cell types composing six whole LN sections and 36 duplicate 2 mm tissue cores from 19 LN (S1 Table), for a total of 7,500,552 cells (5,891,960 cells in the six whole sections), we stained them in saturating conditions [11] with 78 antibodies according to the MILAN method [12] (Fig 1, S1 Fig. and S2 Table). We then developed and applied an analytical pipeline, BRAQUE, consisting of a bundled segmentation algorithm (CyBorgh), a preprocessing phase to enhance cluster separation (called Lognormal Shrinkage), a dimensionality reduction algorithm (UMAP), and a clustering algorithm (HDBSCAN) [10].

BRAQUE is able to identify clusters composed of as little as 12 cells and comprising down to 0.01% of total sample cells (S3 Table); however for maximal detail

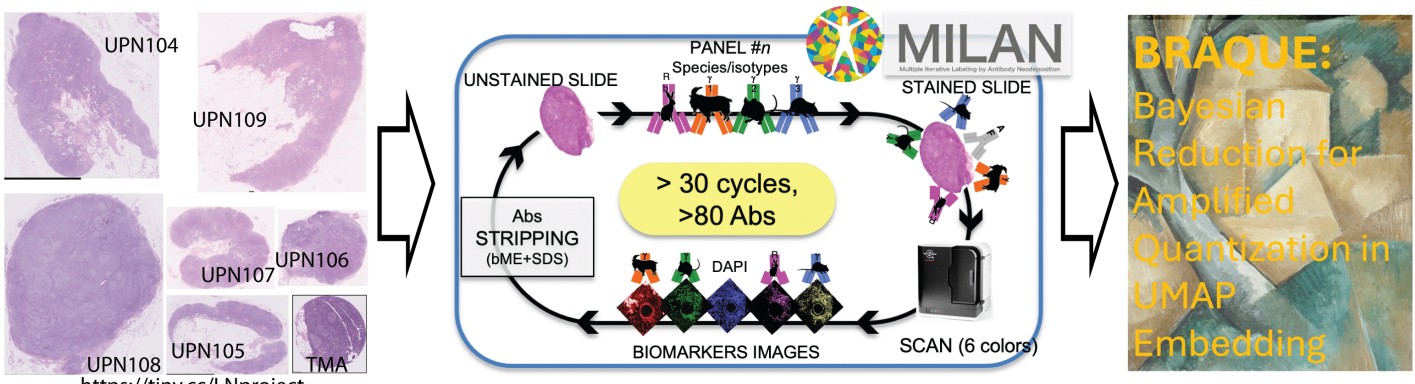

**Fig 1. Global single cell classification by BRAQUE in tissue sections.** Scheme of the BRAQUE analysis pipeline. Right: sections of whole LN H&E images (WSI; scale bar = 5 mm. The TMA core, black image border, has a diameter of 2 mm). The images can be examined at the link to https://tiny.cc/LNproject. The cartoon at the center depicts the cyclic staining and stripping MILAN method, followed by BRAQUE.

we applied a two-step, nested application of BRAQUE to the segmented cells, i.e., a comprehensive BRAQUE$^{global}$ classification (Fig 2, S3 Table, S4 Table, S5 Table) followed by a broad cell-type targeted BRAQUE$^{subclassification}$ analyzing separately CD4 and CD8 T-cells, B-cells, myeloid cells and non-myeloid dendritic cells (Fig 2) (S6 Table). In total, this pipeline produced a 77 cell type classification of the 6 human whole LN samples (S7 Table).

These phenotypic cell types (interchangeably named phenotypic "subsets") are illustrated below.

## T-cells, CD4 and CD8

The cell fate of mature T-cells is dictated by the presence and activity of TCF7, a transcription factor (TF) that provides stem cell-like properties and flexible adaptation [13,14]. TCF7 levels correspond to a time-dependent sequence of activation and resolution of an immune challenge [15], thus we arbitrarily divided T-cells stratified by their TCF7 expression into high, low and average levels (S2 Fig.).

No consistent phenotype was produced by classifying T-cells based on TCF7 expression alone (Table 1); therefore we used the expression levels of TCF7, the presence of FOXP3 (regulatory T-cells) and the co-expression of other markers (checkpoint inhibitors TIM3, PD1, OX40/CD134 and CD137; proliferation indicator Ki-67; activation markers IRF4, HLADR, CD69 etc.) as a strategy to classify clusters of CD4 and CD8 cells (Table 2 and Table 3, S8 Table) (S2 Fig. A,B), that resulted in the identification of 14 CD4 and 13 CD8 cell types (Figs 3, 4A and C, S3 Fig.). The random selection of patient cases, for which any recent antigenic exposure was unknown, was reflected in the heterogeneity in T-cell subset composition in each LN (Fig 4B and D, Fig S3), as well as for any other cell type.

We found that among CD4 + T-cells, FOXP3 and TCF7$^{hi}$ cells were mutually exclusive and T$_{regs}$ were about equally split between having TCF7$^{average}$ and TCF7$^{low}$ (Fig 3). This was expected, given the suppressive effect of TCF7 on the FOXP3 gene expression [29]

In contrast, CD8 + T$_{reg}$ never displayed low TCF7 levels, except when activated.

A portion of T$_{regs}$, both CD4 and CD8, co-expressed FOXP3 and ID2 (Table 2 and Table 3, S8 Table), possible evidence of plasticity [21]. FOXP3 + CD4 and CD8 cells could be further subdivided based on the presence of additional activation markers and PRDM1 [22] (Table 2 and Table 3, S8 Table)(S2 Fig.B). Proliferating and S100B + T$_{reg}$s were also heterogeneous in amount and distribution (Fig 4).

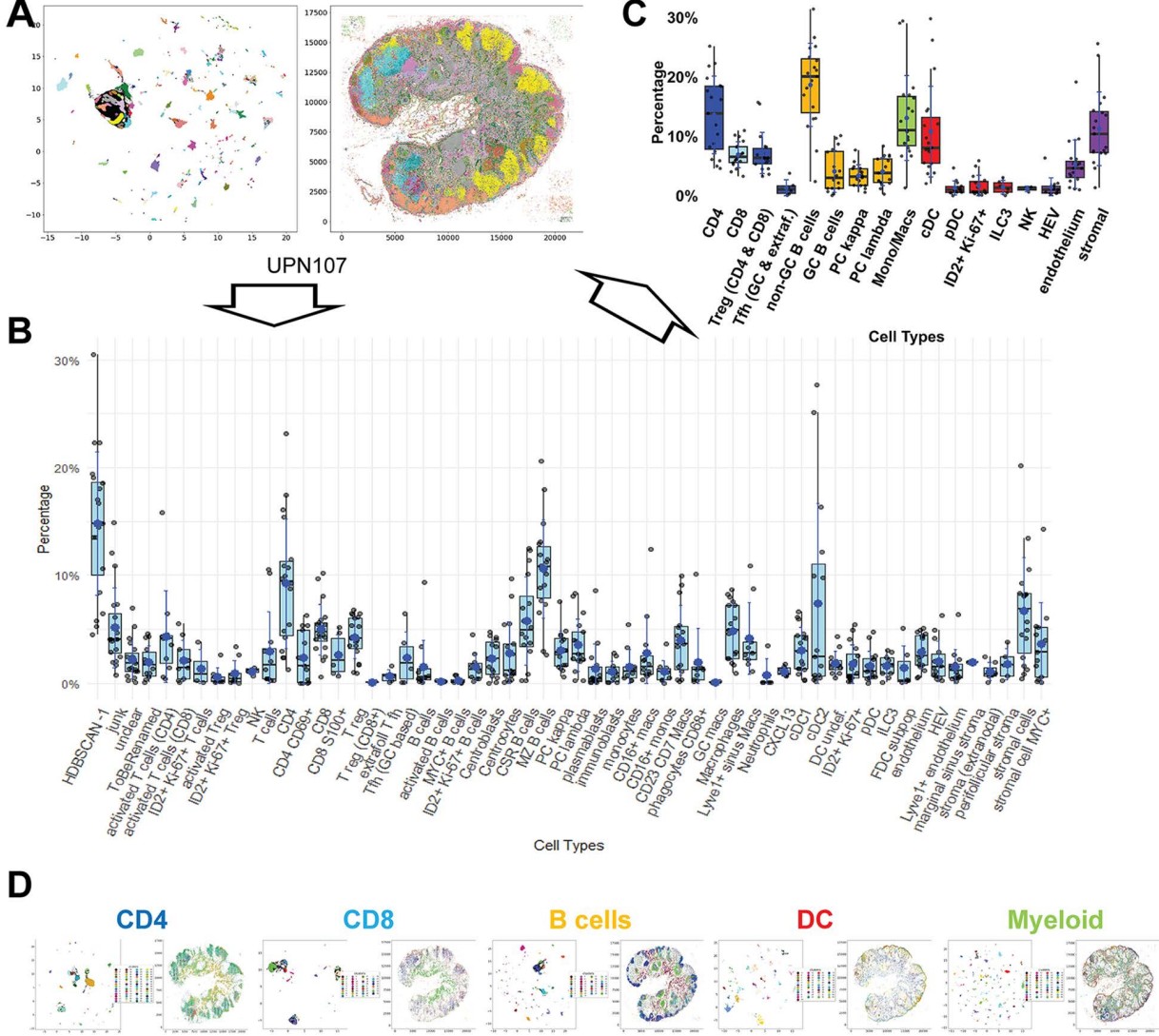

**Fig 2. Global cell classification of human lymph nodes. A:** The UMAP virtual 2D map (left) and the real spatial distribution (right) clustered with BRAQUE^global, is shown for case UPN107 as an example. Each cluster is color-coded. The scale on the left is virtual, the scale on the right are pixels (0.45μm/pixels). **B:** The cell types identified in 27 LN samples (6 whole LN and 21 2mm TMA cores) after BRAQUE^global analysis on all cells in each sample, expressed as boxplots delimited by interquantile ranges, containing the median value (bar) and the mode (asterisk), whiskers indicating min/max values and individual sample points, including outliers. Data are reported in <u>S3 Table</u>. Cells defined as "T cells" and "B cells" cannot be further subdivided into subsets (see <u>Table 1</u>, <u>S5 Table</u> for definitions). **C:** The cell types identified in **B** are grouped as broad cell types, expressed as boxplots (see specification in <u>Fig 2B</u>), excluded HDBSCAN −1, junk and unclassified from the calculation. Data are reported in <u>S3 Table</u>. **D:** The broad categories of CD4, CD8, B cells, dendritic and non-myeloid innate cells (DC) and myeloid cells shown in color in **C** was each separately analyzed by BRAQUE^subclass and each UMAP shown for UPN107 as an example. Images specifications are as per **A**.

The FOXP3-negative T-cells displayed a remarkable mirroring of phenotypic subtypes in CD4 and CD8 T-cells (<u>Table 2</u> and <u>Table 3</u>, <u>S8 Table</u>). Both CD4+ and CD8+T-cells contained TCF7^hi cells expressing progenitor-associated markers (BCL6, GATA3, ID2 and PRDM1) [30,31].

Both subsets also showed TCF7^hi cells with heterogeneous presence of checkpoint inhibitors listed above, cytotoxicity (GZMB) and activation markers (CD23); we defined these cells as being "poised" to acquire an effector phenotype [13,32]

**Table 1. Description, phenotype, numerosity of the T cell types identified by TCF7 values in six whole LN.**

| | Markers frequency in cell-type defined clusters | | | Markers ranked 1st-5th | n. cells | n. clusters |
|---|---|---|---|---|---|---|
| | 70-100% | 50-69% | 30-49% | | | |
| **CD4** | | | | | | |
| TCF7lo | | CD86, FOXP3, IRF4, PRDM1, TIM3, TOX1 | BCL6, CD23, CD30, CD56, CD69, CD137, cMAF, GATA3, GZMB, ID2, MX1, MYC, OX40, PD1 | **FOXP3** | 98871 | 80 |
| TCF7hi | TCF7 | GATA3 | BCL6, CD30, CD69, CD137, GZMB, ID2, MYC, PD1, PRDM1 | CD86, Ki-67, S100B, TCF7 | 263772 | 50 |
| TCF7avg | | CD69, TIM3 | CD23, CD56, CD86, cMAF, CXCL13, GATA3, ID2, IRF4, MX1, MYC, OX40, PD1, PDL1, PRDM1, TOX1 | FOXP3, HLADR | 331053 | 158 |
| **CD8** | | | | | | |
| TCF7lo | | | CD69, CD74, GZMB, PD1, TIM3, TOX1 | CD103, cMAF, EOMES, S100B | 35068 | 58 |
| TCF7hi | TCF7 | | BCL6, CXCL13, GATA3, MYC, PRDM1 | CD56, CD86, cMAF, RORC, TCF7 | 31864 | 46 |
| TCF7avg | | | CD86, CXCL13, OX40, PRDM1, TIM3 | HLADR | 92313 | 92 |

Table 1 legend: Columns 2nd to 4th show the marker frequency (by statistical relevance and by markers ranking) among clusters (i.e., marker-positive clusters/total clusters of that cell type), column 5th the top five markers with greater effect size. Highlighted in bold in column 5th are the markers present in at least 30% of the cells of that type.

or at a stage before transitioning to a TCF7$^{low}$ phenotype [15,17]. Resident memory CD4 T-cells (T$_{rm}$) may be the closest subset described [33].

We defined T-cells with average levels of TCF7 (lacking either high or low levels of this TF), displaying checkpoint inhibitors and activation markers as "effectors", also known as progenitors exhausted (T$_{pex}$; [17]). They could be distinguished from TCF7$^{low}$ cells, the latter bearing a more consistent phenotype (GZMB, CD137, PD1 etc.) and corresponding to "exhausted" T-cells [13].

Both CD4+ and CD8+T-cells contained S100B+ subpopulations, a known naïve subset for CD8+T-cells [26,34]. In addition, S100B+CD8+T-cells with checkpoint inhibition and activation markers could be distinguished from naïve cells and had variable TCF7 levels.

S100B+CD4+subsets with an effector phenotype localized close to CD5+conventional dendritic cells type 2 (cDC2) (Table 2, S8 Table, S9 Table and S10 Table). We could not define if this phenotype was due to a cell autonomous S100B+ phenotype [26] or because of proximity with a S100B+DC with dendrites. We favor the first scenario because the phenotype of these T-cells did not completely match the repertoire of the neighboring DC.

Germinal Center (GC) based T follicular helper CD4+T-cells (T$_{fh}$) clustered separately from the other CD4 cells and displayed the expected phenotypes (Table 2, S8 Table). Remarkably, more than 50% of T$_{fh}$ clusters, together with proliferating and activated T$_{regs}$ and IFN-responsive CD8, contained cMAF, a TF crucial for T$_{fh}$ differentiation [23].

Additionally, CD4+ and CD8+T-cell subsets expressed MX1, a marker of interferon (IFN) signaling [4,19,35], had a mixed TCF7 profile but few other characterizing markers (Table 2 and Table 3, S8 Table) (Fig 5C, S3 Fig.): we defined them as "IFN response" T-cells [4,18].

A population of CD8 T-cells featuring low TCF7 levels and few other characterizing markers (CD8 TCF7$^{lo}$) was found to colocalize and interact with CD8+MX1+cMAF+IFN response T-cells (Fig 5C) (Table 3, S8 Table, S9 Table and S10 Table), suggesting a close functional relationship.

Proximity and likelihood of interaction of cell subsets were assessed using the novel Neighborhood and Overlap analyses. Briefly, statistically significant neighbors are computed for each cell subset using an odds ratio assessment

**Table 2. Description, tissue location, phenotype, numerosity of the CD4 cell types identified in five whole LN.**

| Cell type | Description | Location (% of cells) | Markers frequency in cell-type defined clusters (underscored >70% of cells) | | | Markers ranked 1st-5th (bold: >30% of clusters) | n. cells | n. clusters | UBERON | Othogonal validation Refs. |
| | | | 70-100% | 50-69% | 30-49% | | | | | |
|---|---|---|---|---|---|---|---|---|---|---|
| **CD4** | | | | | | | | | CL:0000084 | |
| CD4 naïve-like | TCF7hi naïve CD4 | paracortex (50–70%); | TCF7 | GATA3, MYC | BCL6, CD30, cMAF, ID2, PRDM1 | CD69, CD74, CD86, CXCL13, GATA3, IRF4, PD1, PDL1, S100B, TCF7, TOX1 | 145,523 | 25 | CL:0000895 | [16,17] |
| CD4 poised | TCF7hi CD4 with co-inhibitory receptors | paracortex (≥70%); | GATA3, GZMB, TCF7 | CD137, MYC, TIM3 | BCL6, CD30, CD69, CD86, LAG3, OX40, PD1, PDL1, PRDM1 | CD86, IRF4, TCF7 | 129,010 | 24 | | [17,18] |
| CD4 proliferating | Proliferating CD4 | interstitial, paracortex, medullary cords (20–50%) | ID2, Ki-67, TOX1 | BCL6, IRF4, MX1, MYC, PRDM1 | CD56, CD69, CD86, CXCL13 | **CD69, ID2, Ki-67** | 13,484 | 14 | | [16] |
| CD4 effector | CD4 with activation and co-inhibitory receptors, TCF7avg (not low) | paracortex, medullary cords (20–50%) | | CD56, CD69, OX40, PD1, TIM3 | CD23, CD86, CD137, cMAF, CXCL13, GZMB, IRF4, MYC, PDL1, PRDM1, TOX1 | **CD86** | 127,683 | 55 | | [17] |
| CD4 effector S100B+ | S100B+CD4 with activation and co-inhibitory receptors | paracortex (≥70%); | CD56, CD69, OX40, PDL1, S100B, TIM3 | CD23, CD103, HLADR, IRF4, PD1 | CD86, cMAF, CXCL13, MYC | **HLADR** | 36,117 | 30 | | |
| CD4 exhausted | TCF7low CD4 with multiple exhaustion markers | medullary cords, Activation spot (20–50%) May be focal | CD56, CD86, CD103, GZMB, MYC, OX40, PD1 | CD137, LAG3, MX1 | CD23, CD30, CD69, CD74, CXCL13, EOMES, PRDM1, RORC, TIM3, TOX1 | CD23, CD69, GZMB, IRF4 | 17,193 | 15 | | [17] |
| CD4 IFN response | MX1+CD4 | paracortex (≥70%); May be focal | MX1 | | CD56, CXCL13, TCF7 | CD69, CD74, CD86, cMAF, CXCL13, GATA3, MX1, OX40, TCF7, TIM3 | 44,546 | 15 | | [18,19] |
| CD4 Treg resting | FOXP3+Regulatory CD4; also named "central" Tregs | interstitial (≥70%); | FOXP3 | CD30, MX1, TIM3, TOX1 | BCL6, CD56, CD86, cMAF, GATA3, ID2, PRDM1 | CD30, CXCL13, FOXP3, PDL1, PRDM1 | 50,180 | 16 | CL:0000815 | [20,21] |
| CD4 Treg activated | FOXP3+Regulatory CD4 with activation or co-inhibitory receptors; also named "effectors" Tregs | interstitial (≥70%); | FOXP3, GATA3, IRF4, PRDM1, TIM3 | CD69, CD86, cMAF, OX40, PD1, TOX1 | BCL6, CD23, CD30, CD56, GZMB, ID2, MX1, MYC, PDL1 | FOXP3, S100B | 96,405 | 41 | CL:0000815 | [20–22] |
| CD4 Treg S100B+ activated | FOXP3+S100B+ Regulatory CD4 with activation/ co-inhibitory recept. | paracortex (≥70%); | CD69, FOXP3, IRF4, OX40, PDL1, PRDM1, S100B, TIM3 | GATA3, HLADR | CD23, CD56, CD86, cMAF, MX1, TOX1 | **FOXP3, HLADR** | 6,403 | 17 | CL:0000815 | [22] |
| CD4 Treg proliferating | Proliferating FOXP3+Regulatory CD4 | paracortex (50–70%); interstitial (20–50%) | FOXP3, GATA3, ID2, IRF4, Ki-67, PRDM1, TOX1 | cMAF, MX1 | BCL6, CD86, MYC, TIM3 | **FOXP3, Ki-67** | 6,735 | 19 | CL:0000815 | [21,23] |

*(Continued)*

**Table 2.** (Continued)

| Cell type | Description | Location (% of cells) | Markers frequency in cell-type defined clusters (underscored >70% of cells) | | | Markers ranked 1st-5th (bold: >30% of clusters) | n. cells | n. clusters | UBERON | Othogonal validation Refs. |
| | | | 70-100% | 50-69% | 30-49% | | | | | |
|---|---|---|---|---|---|---|---|---|---|---|
| CD4 Tfh | Follicular T helper cells | follicular (≥70%); | BCL6, CD56, CD69, CD137, CXCL13, ID2, Ki-67, PD1, TOX1 | _cMAF_ | CD30, CD86, MX1 | **BCL6, PD1, TOX1** | 6,818 | 9 | CL:0002038 | [16,23] |
| CD4 undefined | CD4 of uncertain phenotype | interstitial, paracortex, regional distr. (20–50%) | | | CD30, cMAF | BCL6, CD69, CD74, cMAF, CXCL13 | 13,599 | 8 | | |
| Average CD4 | Default assignment by BRAQUE | NA (≥70%); | _MX1_ | TCF7, TOX1 | CD30, CD56, cMAF, CXCL13, GATA3, GZMB, ID2, IRF4, Ki-67, PD1, TIM3 | CD86, cMAF, FOXP3, PDL1, TCF7 | 66,918 | 4 | | |

Table 2 legend: Columns 4th to 6th show the marker frequency (by statistical relevance and by markers ranking) among clusters (i.e., marker-positive clusters/total clusters of that cell type), column 7th the top five markers with greater effect size. Underlined in columns 4th to 6th are markers expressed in ≥70% of the cells of each cell type (i.e., total of cells positive for that biomarker/total cell of that cell type). Highlighted in bold in column 7th are the markers present in at least 30% of the cells of that type. The last column on the right contains the literature references with orthogonal validation of the phenotype. An expanded version with additional data (list of neighboring cell types and overlapping cells) is available as Supplementary S8 Table.

of a cell type's location within/without a defined distance from another and the significance determined using the Fisher statistical test. With the more stringent Overlap analysis, the likelihood of a cell type being within a defined distance to cells of another type is expressed as a cellular overlap scale in which higher scores indicate a greater propensity for proximity. These analyses and their application to a broader collection of cell types are described in more detail further below.

## B-cells and plasma cells

BRAQUE[superscript subclass] analysis produced 19 B-cell types, the spatial distribution of which landmarks the architecture of LNs without pathology. Some of the identified populations are phenotypically novel (Fig 6, S4 Fig.A).The markers used to classify each cell subset are listed in Table 4 and S8 Table, and include IgD and CD27 to distinguish non-GC mature B cells and memory B cells (MBC) and those that have isotype switched (class switched recombined; CSR) [45,40].

The phenotype and physical location of some subsets are noteworthy. Whereas CD27[neg] B-cells, IgD+ or IgD[neg], had a follicular distribution, populating the Mantle Zone (MZ), all CD27 + B-cells had an almost completely mutually exclusive distribution and were located in the medullary cords or marginal to the B-cell follicles (Fig 7, S5 Fig.).

CD5 expression defined 4 additional B-cell types, including one found in the periphery of the MZ and spilled into the paracortex(CD5+MZ) (Table 4) (Fig 7C, S4 Fig.A). CD5+T-zone B-cells instead were mainly confined within the T-cell zone (Fig 7C, S4 Fig.A) (Table 4, S8 Table). The phenotype of these two CD5+subsets was notable for the absence/low levels of restricted B-cell markers such as CD20 or CD79a (except IgD in 30–50% of CD5+MZ clusters) and a TCF7[low], PAX5[low] phenotype (S4 Fig.A) (Table 4, S8 Table, S11 Table). Phenotypically, these clusters resemble the "transitional naïve B-cells" [38] or T1 B-cells [37,46], i.e., IgD±, CD27[neg], CD43[neg], CD23[neg], rather than B1 cells (CD27+, CD43+,

**Table 3. Description, tissue location, phenotype, numerosity of the CD8 cell types identified in five whole LN.**

| Cell type | Description | Location (% of cells) | Markers frequency in cell-type defined clusters (underscored >70% of cells) | | | Markers ranked 1st-5th (bold: >30% of clusters) | n. cells | n. clus-ters | UBERON | Othogonal vali-dation Refs. |
|---|---|---|---|---|---|---|---|---|---|---|
| | | | 70-100% | 50-69% | 30-49% | | | | | |
| **CD8** | | | | | | | | | CL:0000084 | |
| CD8 naïve-like | TCF7hi/avg Naïve CD8 | interstitial (50–70%); paracortex (20–50%) | | BCL6, TCF7 | CD69, cMAF, CXCL13, GATA3, PRDM1 | BCL6, CD16, CD69, CD86, cMAF, CXCL13, RORC, TCF7 | 45,890 | 42 | CL:0000900 | [14,17,24] |
| CD8 poised | TCF7hi Naïve CD8 with co-inhibitory receptors; also named Tresident memory | interstitial (50–70%); paracortex (20–50%) | CD23, MYC, PD1, PRDM1, TCF7 | CD137, CXCL13, GATA3, GZMB, OX40 | BCL6, CD69, CD86, ID2, LAG3, TIM3 | CD56, CD74, CD86, cMAF, PD1, TCF7, TOX1 | 6,993 | 10 | | [14,16,17,24–25] |
| CD8 S100B+ naïve like | S100B+TCF7hi Naïve CD8 | interstitial (≥70%); | | BCL6, CXCL13, MYC, OX40, PD1, PRDM1, S100B, TCF7 | | **MYC, S100B, TCF7** | 3,171 | 2 | | [26,27] |
| CD8 S100B+ other | S100B+CD8 with co-inhibitory recep-tors; any TCF7, any FOXP3 | interstitial (≥70%); | CD86, S100B, TIM3 | FOXP3, GATA3, HLADR, IRF4, MYC, OX40, PDL1, PRDM1, TCF7 | BCL6, CD23, CD56, CD137, CXCL13, GZMB, ID2, LAG3 | CD86, EOMES, HLADR, S100B | 16,246 | 10 | | [26,27] |
| CD8 proliferating | Proliferating CD8, any TCF7 value | interstitial (50–70%); medullary cords (20–50%) | Ki-67 | CD86, EOMES, ID2, IRF4, PRDM1, TIM3 | BCL6, GATA3, MX1, MYC, TOX1 | BCL6, CD30, CD103, EOMES, ID2, Ki-67 | 4,361 | 12 | | [16,24] |
| CD8 effector | CD8 with co-inhibitory receptors, TCF7avg (not hi, neither low); also named Progenitor Exhausted | interstitial, medullary cords (20–50%) | | CD23, CD137, GZMB, OX40, PD1, PDL1, PRDM1, TIM3 | CD16, CD56, CD74, CD86, ID2, MYC | HLADR, IRF4, LAG3 | 34,312 | 29 | | [14,16,17,24,25] |
| CD8 exhausted | TCF7low CD8 with multiple co-inhibitory receptors | interstitial (50–70%); medullary cords (20–50%) | GZMB | CD23, PD1 | CD69, CD74, CD137, MYC, OX40, TIM3, TOX1 | CD74, CD103, GZMB, PD1, S100B, TIM3 | 18,715 | 23 | CL:0000910 | [14,24] |
| CD8 IFN response | MX1+CD8, any TCF7 value | interstitial, capsular (20–50%) | MX1 | cMAF | CD74, CD86, CXCL13, TIM3 | BCL6, CD69, CD74, CD86, cMAF, CXCL13, EOMES, MX1 | 10,807 | 13 | | [16] |
| CD8 TCF7lo | TCF7low with a naïve phenotype | interstitial, paracortex (20–50%) | | CD69, CXCL13 | cMAF, TIM3 | BCL6, CD69, CD74, CD86, cMAF, CXCL13, EOMES, ID2, PDL1, TOX1 | 3,776 | 16 | | |
| CD8 Treg resting | FOXP3+TCF7hi/avg Regulatory CD8, resting | paracortex (50–70%); interstitial (20–50%) | FOXP3 | CD69, GATA3, PRDM1, TCF7 | CXCL13, LAG3, TOX1 | **FOXP3, GATA3, PRDM1, TCF7** | 4,239 | 6 | | [28] |
| CD8 Treg activated | FOXP3+Regulatory CD8 with activation | interstitial (≥70%); paracortex (20–50%) | FOXP3, TIM3 | CD69, CD86, GATA3, IRF4, PD1, PRDM1, TOX1 | CD74, CD137, ID2, LAG3, MX1 | CD56, CD137, EOMES, FOXP3 | 4,172 | 8 | | [22,28] |

*(Continued)*

**Table 3.** (Continued)

| Cell type | Description | Location (% of cells) | Markers frequency in cell-type defined clusters (underscored >70% of cells) | | | Markers ranked 1st-5th (bold: >30% of clusters) | n. cells | n. clusters | UBERON | Othogonal validation Refs. |
|---|---|---|---|---|---|---|---|---|---|---|
| | | | 70-100% | 50-69% | 30-49% | | | | | |
| CD8 undefined | CD8 of uncertain phenotype | medullary cords (50–70%); interstitial (20–50%) | | | | CD86 | 6,563 | 25 | | |
| Average CD8 | Default assignment by BRAQUE | NA (≥70%); | | CXCL13, MX1 | OX40, PD1, PDL1, TIM3 | CD69, CD86, MX1, PRDM1, S100B, TCF7, TOX1 | 31,120 | 5 | | |

Table 3 legend: See legend to Table 2.

CD5±). CD5+MZ and T-zone cell subsets were identified in about 40% of the LNs examined (S11 Table) and in all cases in the same marginal follicular zone location.

CD5+light chain+ B-cells express high levels of kappa or lambda immunoglobulin light chain that also identifies plasma cells and plasmablasts. These cell types all occupy the medullary cords (Fig 8A, S4 Fig.A) (Table 4, S8 Table). However, CD5+light-chain+ B-cells have a phenotype that is suggestive of the B1 B-cell type [44], occupy a non-identical LN environment compared to CD5$^{neg}$ plasma cells (Fig 8A) and show an independent k/l light chain ratio (Fig 8C).

Non-pathologic, incidentally discovered LN do not have a prominent follicular activation by definition and thus they may be less suited to analyze the GC reaction. In the few well-developed GC, polarization was noted (Fig 7B). Occasional centroblasts and centrocytes were found as isolated cells outside the GC, as has been described in mice LN [47] (S4 Fig.A).

Two additional cell types featured a spatial relationship with the GC.

ID2+Ki-67+ and MYC+B-cells were scattered in much of the LN interstitium, the paracortex, the follicle and entered the dark zone (DZ) and the outer zone of the GC (Fig 7B, S4 Fig.A). ID2+Ki-67+B-cells shared phenotypic traits with centroblasts and they may be part of the extra-GC wave of early B-cell activation [47].

IRF4+PRDM1+GC-based centrocytes were not identified, contrary to the expectation [48] perhaps due to the small size of the GCs sampled and the LN origin [38].

Extrafollicular, activated B-cells termed "immunoblasts" [49], coexpress IRF4, PRDM1 and sporadically CD30 (BRAQUE$^{global}$) but lack other features of plasma cells or MYC expression. This subset was found in two locations: the paracortical T-zone and the medullary cords (Table 4, S8 Table) (S4 Fig.A).

Kappa or lambda light chain expression marks plasma cells, plasmablasts and the CD5+light chain+ B-cells mentioned above. These cells all occupy the medullary cords (Fig 8A, S4 Fig.A) (Table 4, S8 Table).

The ratio of circulating kappa and lambda immunoglobulin light chains in the bloodstream it has been established to be around 2:1 [50]; in the tissue, wider variations of the light chain-bearing B-cell ratios have been reported [51–53]. In our samples the k/l ratio was 1.5 (SD ± 1.13), falling below 1 in several whole LNs and TMA core samples (Fig 8B). We confirmed these data by segmenting and quantifying the plasma cells with traditional image analysis tools (thresholding and particle counting) and found similar values (S6 Fig.). Lastly, we measured the k/l ratio in 7 whole reactive LNs by in-situ light chain hybridization [54], obtaining a 1.38 ± 0.40 ratio (range 0.87–2.17) (Figs S6C, S6D).

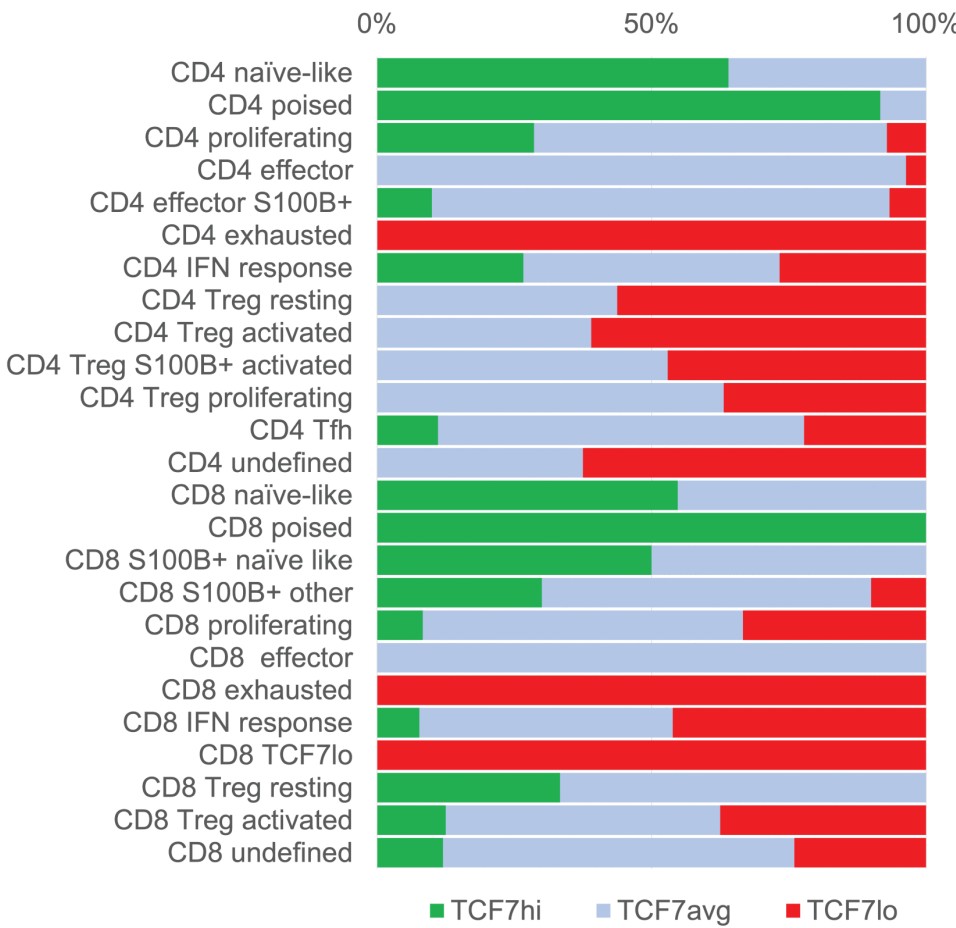

**Fig 3. T-cell classification according to TCF7 levels.** Levels of TCF7 in CD4 and CD8 T-cell subsets. TCF7 levels (high, average or low; see S2 Fig.) are plotted for each CD4 and CD8 cell subtypes as percentage of clusters belonging to a subtype having that level.

## Innate immune and dendritic cells

Fourteen non-myeloid innate immune and DC types were identified by BRAQUE[subclass], none of which expressed CD14 or CD163. cDC1 identified in BRAQUE[global] bore CD14 (S5 Table) as other non-myeloid cells, but nested BRA-QUE[subclass] analysis revealed instead bona-fide monocytes and macrophages not previously identifiable in the DC group (Table 5, S8 Table) (Fig 9, S4 Fig.B). This was unique to DC and not seen upon re-analysis of the other cell types (not shown).

cDC1 is split into two groups: one with the canonical phenotype, cDC1A (CD74, CD141, Clec9A, LYZ, TIM3 in >70% of clusters), preferentially located in the deep paracortex and diffusely, and the other, cDC1B, with more sparse expression of the aforementioned markers (Table 5, S8 Table) (S4 Fig.B), located in the medullary cords [70].

cDC2 is also split in two main subgroups: CD207 + cDC2 (cDC2A, also known as Langerhans cells or LC) and CD207[neg] cDC2 (cDC2B). The main differences were: expression of Langerin/CD207 and CD1a for cDC2A, and a CD4 + CD86 + prevalent phenotype and a predominant location in nodular paracortical aggregates, close to PNA+HEV (Fig 10A) for cDC2B.

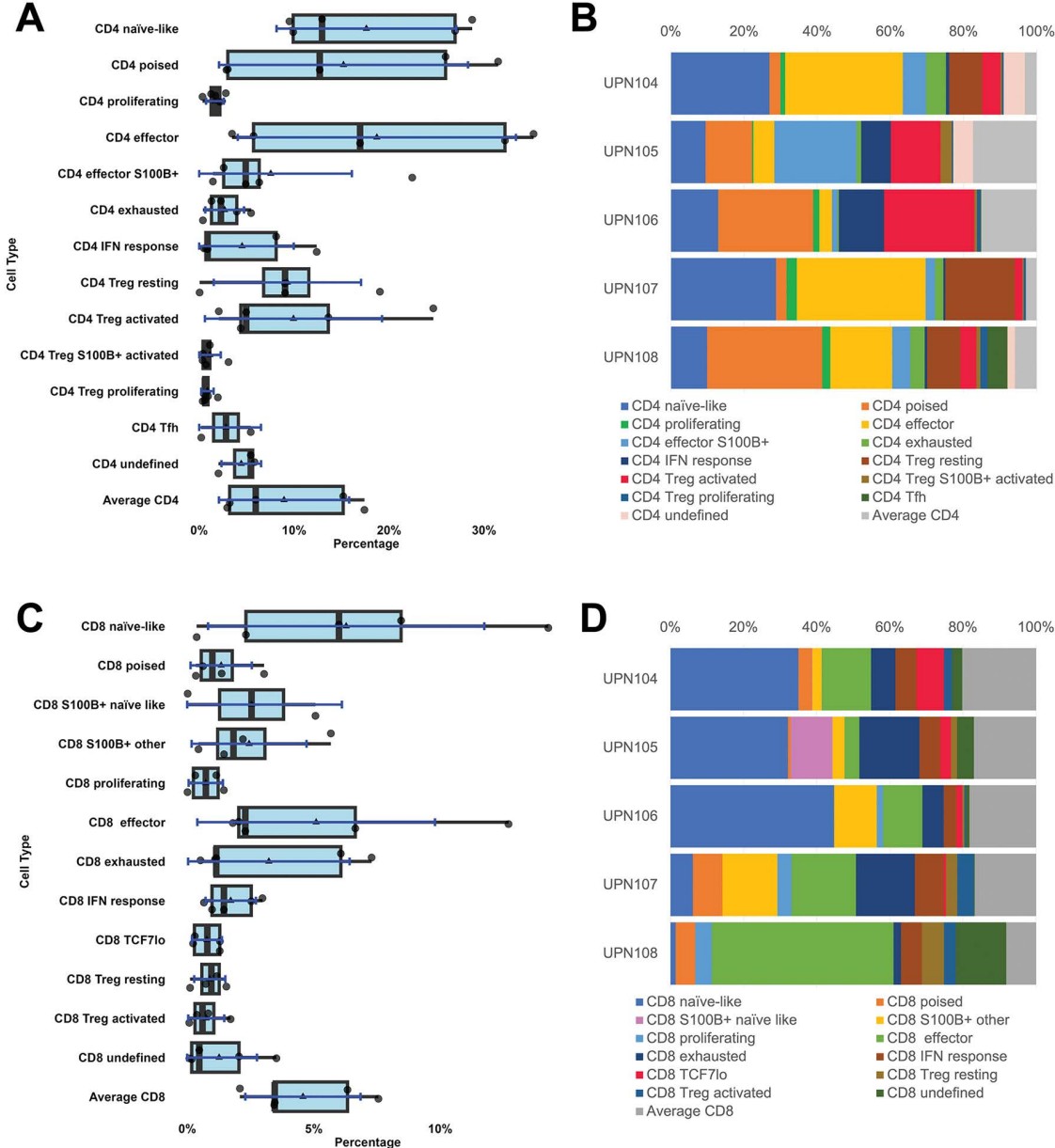

**Fig 4. T-cell classification. A, B:** CD4 T-cells. T-cell types obtained after BRAQUE from five whole LNs represented as boxplots (**A**) and subtype percentage composition for each individual LN (**B**). For the boxplots specifications see Fig 1. **C, D**: CD8 T-cells. T-cell types obtained after BRAQUE from five whole LNs represented as boxplots (**C**) and subtype percentage composition for each individual LN (**D**). For the boxplots specifications see Fig 1.

Neither group could be defined by HLA-DR or CD11c differential expression, however only cDC2A were CD1A+, a phenotype described for resident cDC2 [56].

cDC2A and cDC2B were themselves split into CD5+ and CD5$^{neg}$ subsets, as previously shown in blood [59,55]. Both CD5$^{neg}$ cDC2 subsets express IRF4, PU1 and BCL6, however their spatial localization was superimposable to the CD5+cDC2s (Fig 11A) (S4 Fig.B). Conventional image rendering could not highlight membrane CD5 staining on cDC2, as previously shown [71](Fig 10B).

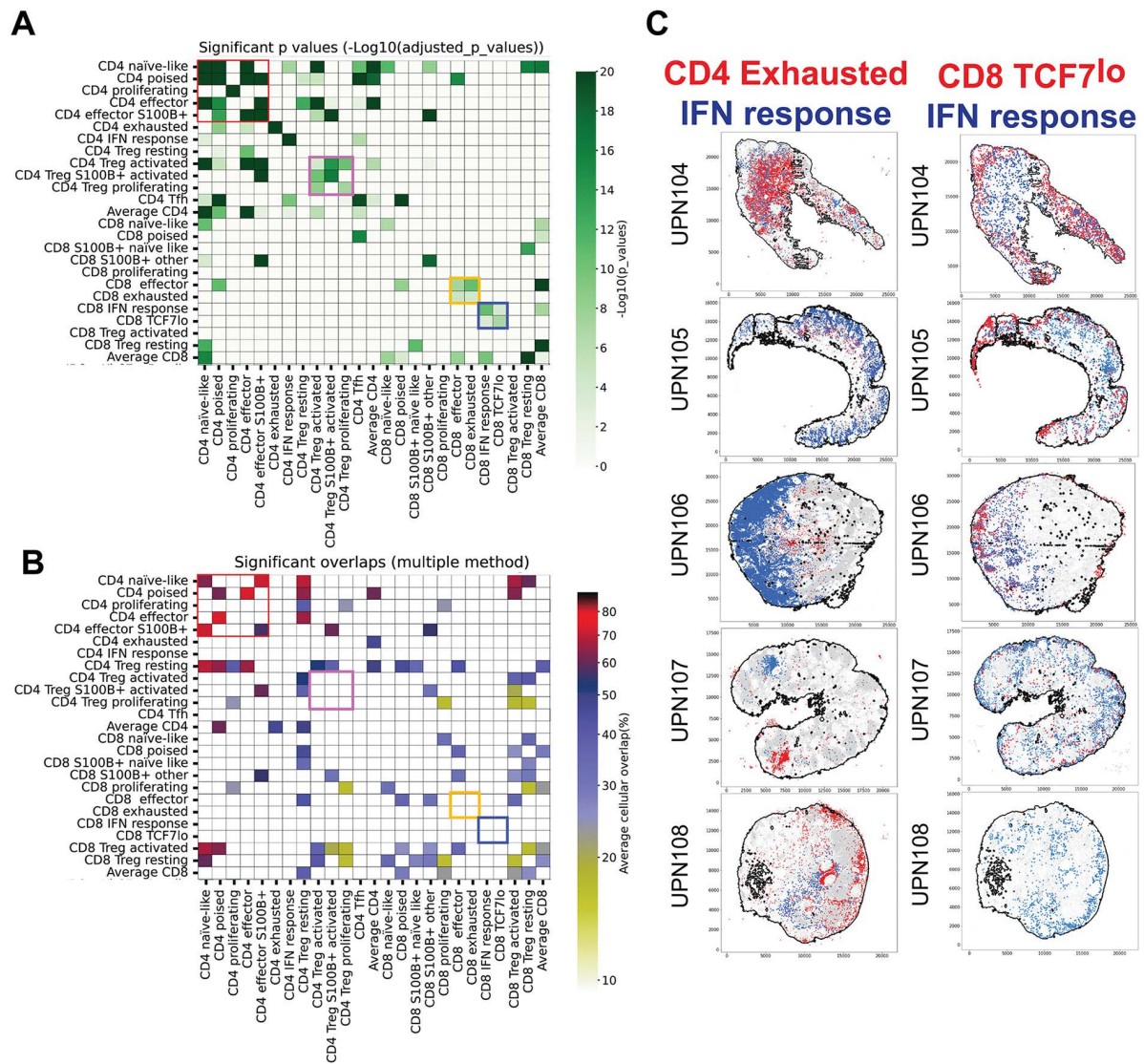

**Fig 5. Spatial relationships of CD4 and CD8 T-cell subsets. A:** CD4 and CD8 neighborhood significant relationships. For each T-cell type the statistically significant neighbors are computed. Only significant values are plotted in shades of green as per the scale on the right. Note that CD4 with high or average TCF7 levels tend to be neighbors (red square). Activated and proliferating CD4 Treg, but not resting Treg also congregate together (pink square). CD8 effectors and exhausted are neighbors (orange square) as two CD8 subsets, CD8 TCF7$^{lo}$ and CD8 IFN signal do (blue square). Average CD4 and CD8 represent HDBSCAN −1 clusters. 518,116 CD4 and 190,365 CD8 cells analyzed. The complete neighborhood and overlap chart can be seen in Fig 14. **B:** CD4 and CD8 subtypes significant overlaps. CD4 with high or average TCF7 levels occupy overlapping spaces in the whole LN (red square); scale on the right. $T_{reg}$ cell space, resting CD4 and activated or resting CD8, overlap with several other T-cell subsets. Note that previously noted (**A**) selected neighborhoods (pink, red and blue squares) do not correspond to spatial overlap regions. **C**: Examples of spatial co-localization of T-cell subsets in whole LN sections. CD4 exhausted (red dots) and CD4 IFN response T-cells (blue dots) are examples of cell types with non-overlapping distribution. CD8 TCF7$^{lo}$ (red dots) and CD8 IFN response T-cells (blue dots) are examples of neighboring cell types. Five individual whole LN are shown.

Cells with a mixed cDC1 and cDC2 phenotype, DC$_{mPh}$ [56,65], had CD86 as the most prevalent marker, a nondescript tissue distribution, did not self-aggregate but were neighbors of cDC2B, S100B+ activated Tregs and putative transitional B-cells (CD5 + T-zone cells) in the paracortex (Fig 11A).

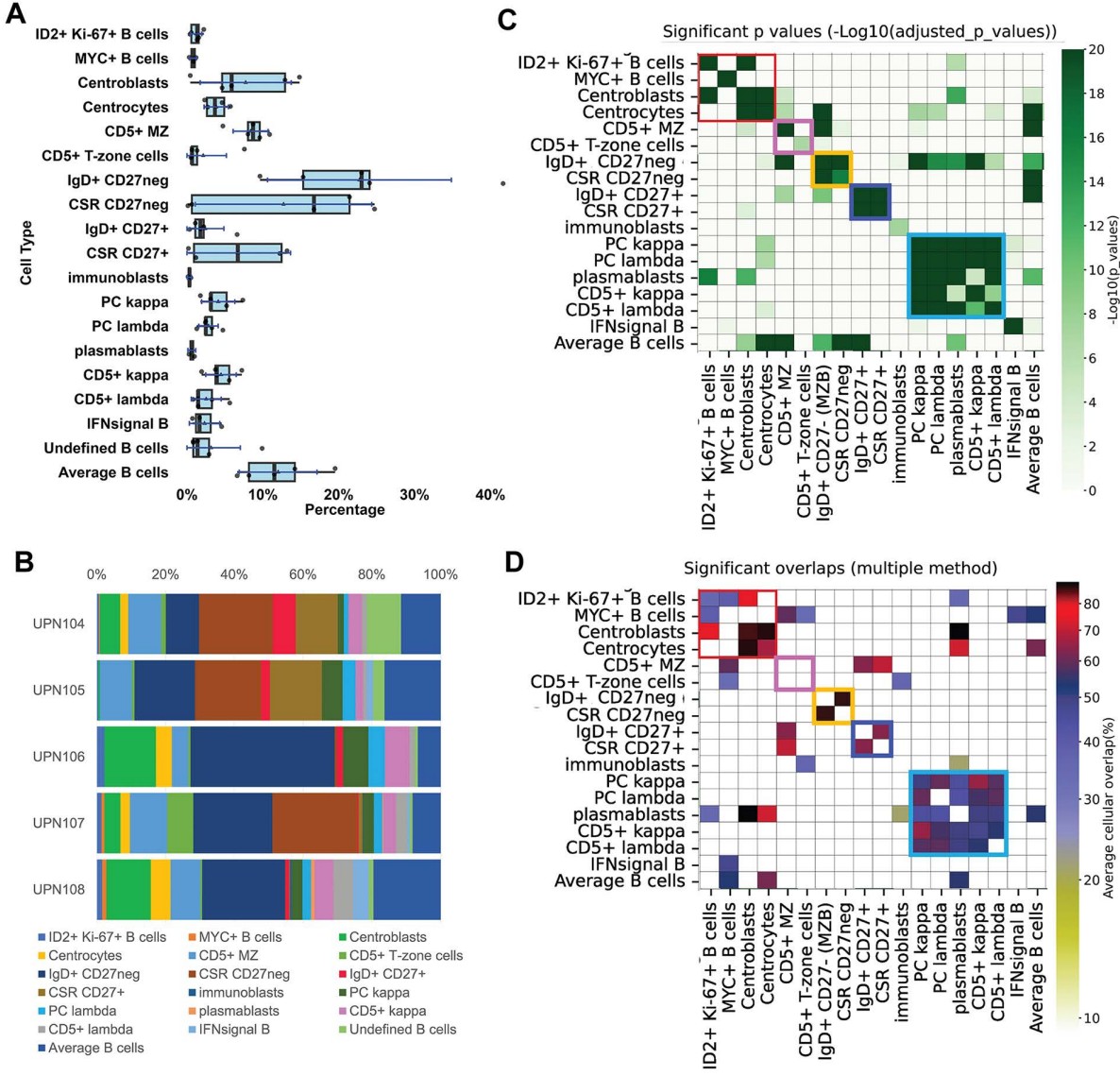

**Fig 6. B-cell classification and spatial distribution. A:** B-cell types from five whole LNs represented as boxplots. For the boxplots specifications see Fig 2. Average B-cells are identified by default. MZ: mantle zone; CSR: class switch recombined. **B:** B-cell subtypes percentage composition of five whole lymph nodes. **C:** B-cell neighborhood significant relationships. For each B-cell type the statistically significant neighbors are computed. Only significant values are plotted in shades of green as per the scale on the right. Note that GC cells, proliferating ID2+ and MYC+ cells are neighbors (red square). CD5+B-cells (kappa and lambda$^{negative}$) form a neighborhood (pink square). CD27$^{negative}$ (yellow square) and CD27$^{positive}$ (purple square) form mutually exclusive neighborhoods. Light chain$^{positive}$ B cells are part of a single neighborhood (blue square). **D:** B cell subtypes significant overlaps. The previously noted neighborhoods (**C**) largely correspond to population spatial overlaps, except for CD5+subsets (pink square).

We identified proliferating ID2+cells lacking T, B or the ILC3-defining marker RORC. The recently published Ki-67+cDC-ILC3 subpopulation [67] may therefore not reside in this fraction, but rather in the proliferating cDC1 fraction. Up to 50% of clusters express AXL, suggesting that tissue-based equivalent of proliferating ASDC [72] are contained in this group.

Plasmacytoid dendritic cells feature a very consistent homogeneous phenotype (CD303, TCF4) and a signature of IFN signaling (MX1) in >70% of clusters, as noticed previously [55] (Table 5, S8 Table) (Fig 11B, S4 Fig.B). GZMB was not

## Table 4. Description, tissue location, phenotype, numerosity of the B cell types identified in six whole LN.

| Cell type | Description | Location (% of cells) | Markers frequency in cell-type defined clusters (underscored >70% of cells) | | | Markers ranked 1st-5th (bold:>30% of clusters) | n. cells | n. clusters | UBERON | Othogonal validation Refs. |
|---|---|---|---|---|---|---|---|---|---|---|
| | | | 70-100% | 50-69% | 30-49% | | | | | |
| **B cells** | | | | | | | | | CL:0000236 | |
| ID2+Ki-67+B cells | ID2+Proliferating B cells | GC+scattered (50–70%); interstitial (20–50%) | BCL6, ID2, Ki-67, TOX1 | AID, CD86, PAX5 | CD10, CD20, CD74, HLADR, IRF8, MYC, PRDM1, PU1, TCF7 | BCL6, CD79a, ID2, Ki-67 | 18,315 | 18 | | [16,36] |
| MYC+B cells | MYC+B cells | interstitial, marginal zone (20–50%) | MYC | CD74, CD86, IRF8, PRDM1, TCF7 | AID, BCL2, BCL6, CD1c, CD5, CD20, CD21, CD79a, Ki-67, PAX5, PDPN, PU1, TOX1 | BCL6, IgD, IRF4, Ki-67, MYC, TOX1 | 19,326 | 9 | | [16] |
| Centroblasts | GC dark-zone proliferating B cells | follicular (≥70%); | AID, BCL6, CD10, CD20, CD21, CD74, CD86, HLADR, ID2, IRF8, Ki-67, PAX5, PDPN, PU1, TOX1 | CD23 | MX1, PRDM1 | **BCL6, CD21, Ki-67** | 96,464 | 91 | CL:0009112 | [16] |
| Centrocytes | GC light-zone B cells | follicular (≥70%); | CD20, CD21, CD23, CD86, PDPN, TOX1 | AID, BCL6, CD10, CD69, CD74, HLADR | CD11c, ID2, IRF8, PAX5, PU1 | CD1c, CD21, CD23, IRF4, PDPN, PRDM1 | 35,677 | 30 | CL:0009111 | [16] |
| CD5+MZ | CD5+mature B cells, coexpressing PAX5 and TCF7 | marginal zone (50–70%); follicular (20–50%) | CD5, TCF7 | BCL2, TOX1 | AID, CD11c, CD23, CD27, CD69, CD74, IRF8, MYC, PDPN, PRDM1, PU1 | BCL2, CD5, CD27, IgD, IRF8, lambda | 125,346 | 38 | | [37–39] |
| CD5+T-zone cells | CD5+TCF7+B cells | paracortex NOS (20–50%) | BCL6, CD5, CD86, PRDM1, TCF7 | IRF4, MYC | CD1c, CD11c, HLADR, ID2, TOX1 | CD1c, ID2, IRF4, MX1, TCF7 | 56,274 | 22 | | [16,37–39] |
| IgD+CD27neg | IgD+ mature B cells residing in the follicular mantle zone | follicular (≥70%); | CD20, CD79a, IgD, PAX5, PU1 | CD1c, CD74, HLADR, MX1 | BCL2, IRF8, PDPN | CD79a, IgD, IRF4 | 384,855 | 122 | CL:0000788 | [16,40,41,42] |
| CSR CD27neg | Mature B cells, IgDneg CD27 neg residing in the follicular mantle zone | follicular (≥70%); | CD74 | BCL2, CD79a, PU1 | CD1c, CD20, CD21, IRF8, MX1, PAX5, PDPN | CD1c, CD11c, CD20, CD21, CD23, IRF8, kappa, lambda, PDPN, PU1 | 139,665 | 42 | CL:0000788 | [40,41–43] |
| IgD+CD27+ | CD27+mature B cells, IgD+ | medullary cords (50–70%) | BCL2, CD20, CD23, CD27, CD79a, IgD | IRF8, MYC, PRDM1, PU1 | CD74 | BCL2, CD21, CD69, CD86, HLADR, IRF8 | 42,066 | 17 | CL:0000787 | [16,40,41,42] |
| CSR CD27+ | CD27+mature B cells, IgDneg | medullary cords (≥70%); | BCL2, CD20, CD23, CD27 | CD5, CD74, CD79a, IRF8, MYC, PRDM1, TOX1 | AID, CD69, PU1 | BCL2, CD21, CD27, CD69, IRF8 | 72,064 | 17 | CL:0000787 | [16,40,41,42] |

*(Continued)*

**Table 4.** (Continued)

| Cell type | Description | Location (% of cells) | Markers frequency in cell-type defined clusters (underscored >70% of cells) | | | Markers ranked 1st-5th (bold:>30% of clusters) | n. cells | n. clusters | UBERON | Othogonal validation Refs. |
|---|---|---|---|---|---|---|---|---|---|---|
| | | | 70-100% | 50-69% | 30-49% | | | | | |
| immunoblasts | Activated mature B cells | interstitial (20–50%) | CD5, CD69, IRF4, MYC, PRDM1 | BCL2, CD27, TCF7 | | CD5, CD30, IRF4, Ki-67, MYC, TCF7 | 4,931 | 6 | | [16] |
| PC kappa | Plasma cells kappa | medullary cords (≥70%); | CD27, CD69, IRF4, kappa, PRDM1 | CD11c, CD23, CD86, ID2, MYC | AID, CD5, CD10, MX1, PDPN, TCF7, TOX1 | **IRF4, kappa, PRDM1** | 63,474 | 30 | CL:0000786 | [16] |
| PC lambda | Plasma cells lambda | medullary cords (≥70%); | CD27, IRF4, lambda, PRDM1 | AID, CD23, CD69 | CD5, CD10, CD11c, ID2, MYC, PDPN, TOX1 | **IRF4, lambda, PRDM1** | 48,072 | 27 | CL:0000786 | [16] |
| plasmablasts | Immature plasma cells | marginal zone, GC + scattered (20–50%) | IRF4, Ki-67, PRDM1 | ID2, kappa, lambda, MX1, MYC | AID, BCL6, CD27, CD86, TOX1 | ID2, IRF4, Ki-67, lambda, PRDM1 | 2,161 | 6 | CL:0000980 | [16,42] |
| CD5 + kappa | CD5 + CD27 + mature B cells, Ig light chain positive | medullary cords (≥70%); | CD5, CD27, kappa, PRDM, TCF7 | CD11c, CD69 | AID, BCL2, CD23, IRF4, MX1, MYC | CD1c, CD27, kappa | 56,728 | 15 | | [16,38,39,42,44] |
| CD5 + lambda | CD5 + CD27 + mature B cells, Ig light chain positive | medullary cords (≥70%); | CD5, CD27, lambda, TCF7 | AID, CD11c, CD69, PRDM1 | CD23, IRF4, MX1, TOX1 | CD1c, CD5, CD27, CD86, IRF8, lambda | 43,704 | 26 | | [16,38,39,42,44] |
| IFNsignal B | Mature B cells with evidence of IFN signaling | marginal/subcapsular z., follicular (20–50%) | MX1 | CD20 | BCL2, CD74, HLADR | BCL2, CD1c, CD10, CD11c, CD20, CD27, CD86, ID2, IRF8, lambda, MX1 | 20,036 | 16 | | [16] |
| Undefined B cells | B cells of uncertain phenotype | marginal zone (20–50%) | | lambda | AID, CD10, CD11c, CD27, CD69, CD86, ID2, IRF4, MYC, PDPN, PRDM1 | CD21, CD74, IRF4, kappa, lambda, PU1 | 35,193 | 28 | | |
| Average B cells | Default assignment by BRAQUE | | CD5, CD11c, MYC | AID, CD23, CD69, HLADR, TOX1 | CD21, CD30, CD74, CD79a, IRF8, lambda, MX1, PDPN, PRDM1, TCF7 | CD11c, CD86, IgD | 179,345 | 6 | | |

Table 4 legend: See legend to Table 2.

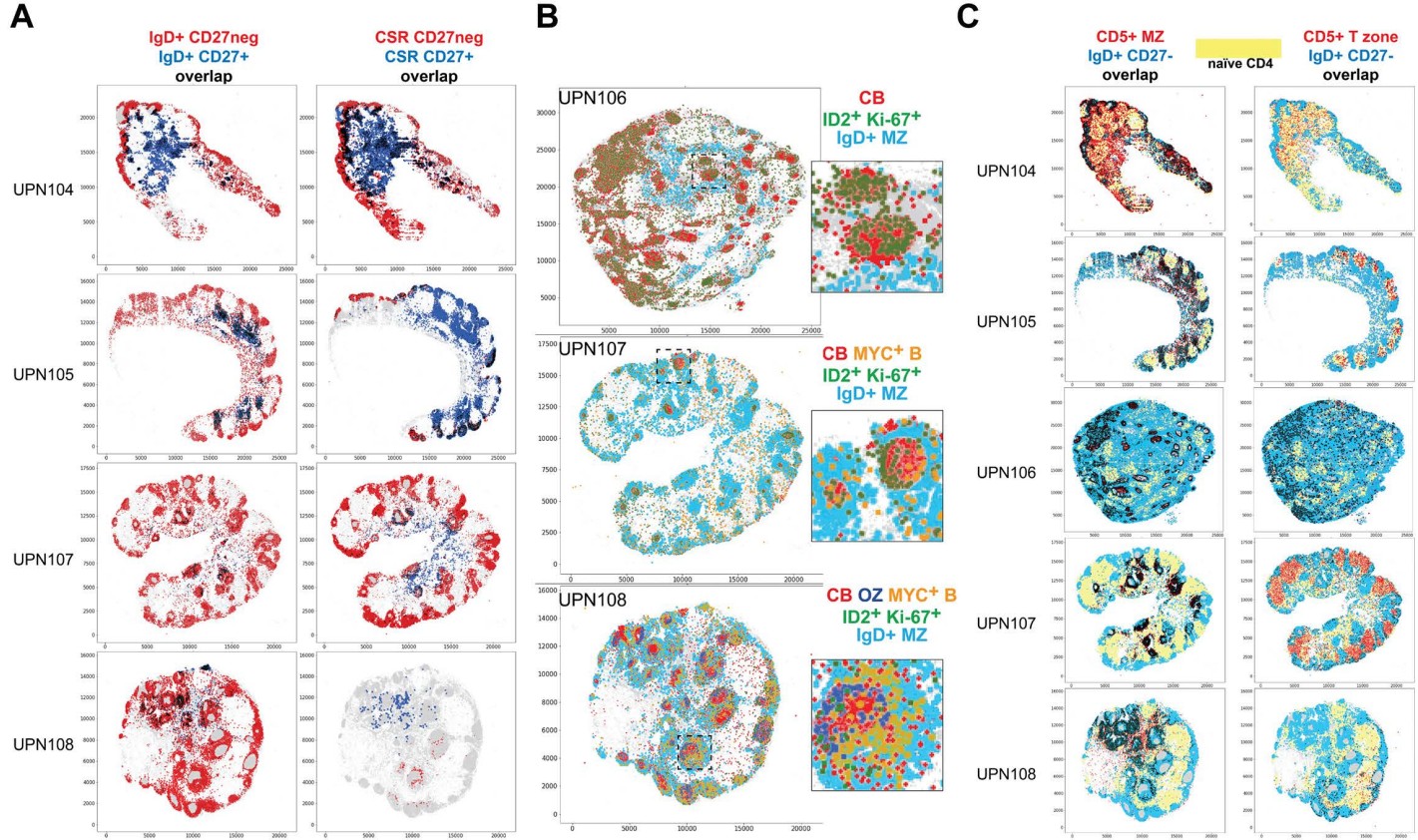

**Fig 7. Tissue landscaping of B-cells subsets. A:** Spatial distribution of mature B-cell types, CD27+ or CD27neg. The spatial distribution of IgD+B-cells and CSR B-cells is shown, divided according to the CD27 expression: negative in red, positive in blue. Overlapping cells are in black. The scale represents pixels (0.45 µm/pixel). **B:** The Germinal Center (GC) environment components are shown for three whole LN. Five cell types are shown: centroblasts (red), proliferating ID2+cells (green), MYC+B-cells (orange), outer zone GC cells (OZ) (blue). The IgD+MZ B-cells are shown to locate the B-cell follicle (turquoise). Note the extra follicular location of ID2+Ki-67+ and MYC+ cells. Centrocytes are not shown. An area of each image is magnified as an inset. The scale represents pixels (0.45 µm/pixel). **C:** Spatial distribution of CD5+B-cell types. The spatial distribution of CD5+MZ and CD5+T-zone cells for five whole LN is shown in red. IgD+CD27- MZ B-cells (blue) and the area occupied by naïve CD4 T-cells (yellow) are shown for spatial reference. Overlapping cells are in black. The scale represents pixels (0.45 µm/pixel). The complete neighborhood and overlap chart can be seen in Fig 14.

detected, as expected in quiescent pDC [73]. pDC aggregates, thought to be typical of this cell type [74], were detected in only 3/6 LN and in a minority of clusters.

ILC3s are described here in situ for the first time in LN; besides RORc, a defining but not entirely unique marker [66,67], ILC3s expresses PU1 and a heterogeneous array of other DC-related cell markers (AXL, CD11c, CD74, TIM3) in >50% of clusters (Table 5, S8 Table). These cells are found in the superficial cortex, next to HEV, as previously noted [75] (Fig 11B) (S4 Fig.B) and also dispersed in the outer T-cell zone, in the medulla and occasionally in the subcapsular sinus.

A single cell type corresponding to NK cells was identified and the phenotype was consistent with the consensus: CD16+CD56+ and others [69]. NK cells were one of the smallest subsets and were often aggregated, close to macrophages (Table 5) (Fig 12C) (S4 Fig.B).

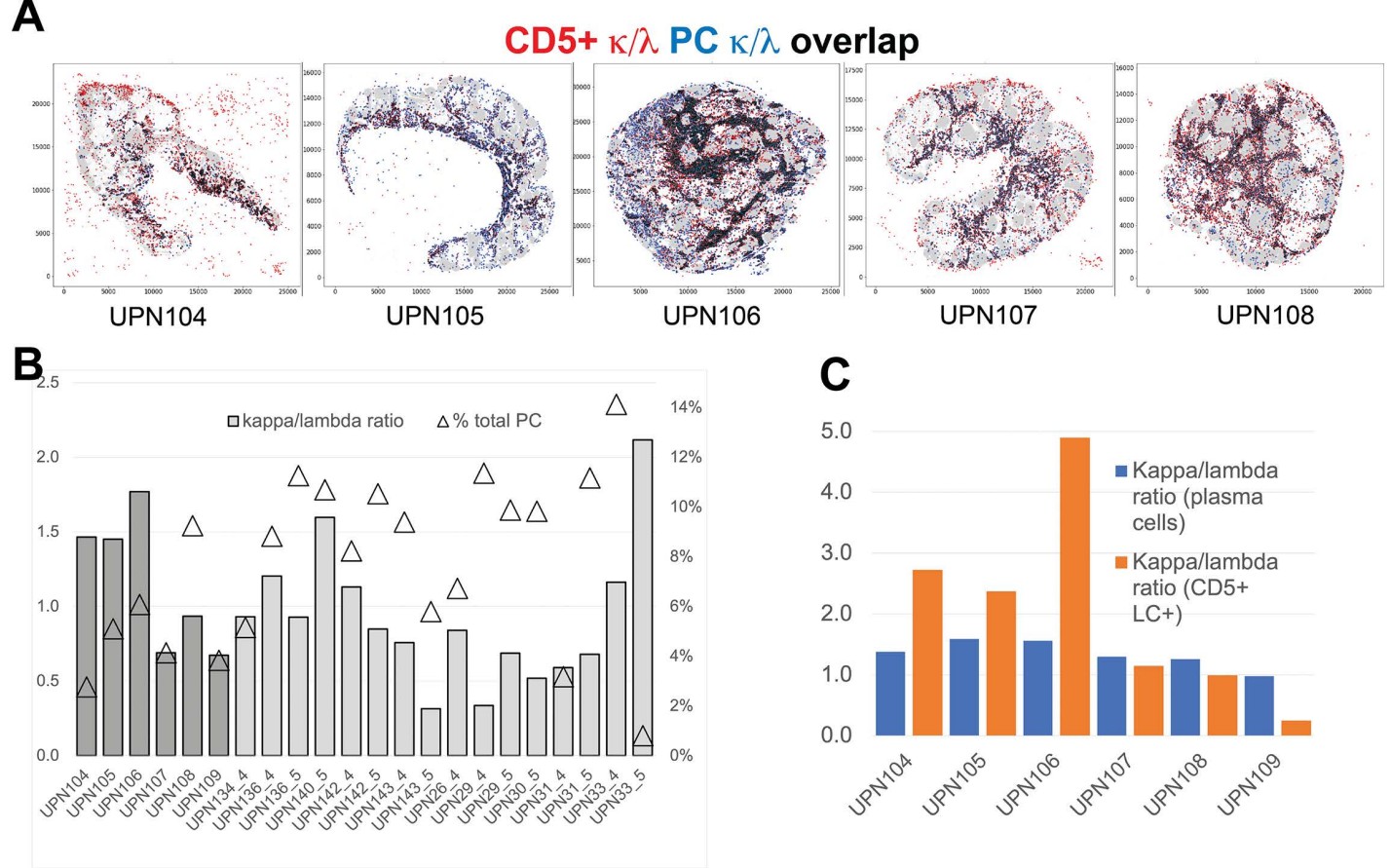

**Fig 8. Spatial distribution and frequency of Ig light chain+ B cells. A:** Spatial relationship between CD5 + Ig light chain+ B-cells and plasma cells. The CD5 + Ig light chain kappa+ and lambda+ (red) are spatially plotted together with the kappa and lambda positive plasma cells (blue). Cell overlap is in black. **B:** Kappa/lambda ratio and plasma cell frequency from BRAQUE$^{global}$. The two-scale graph shows the kappa/lambda plasma cell ratio for 16 LN. The first six (darker columns) are whole LN, the remaining are TMA cores, six of them duplicates. The bars (scale on the left) show the k/l ratio. The triangle (scale on the right) the percentage of PC in each sample. **C:** Comparison of the kappa/lambda ratio in plasma cells and in CD5 + light chain+ (LC) B-cells from BRAQUE$^{subclass}$. The k/l ratio of plasma cells (blue) is compared with the same ratio for CD5 + light chain+ B-cells (orange) in six whole LN.

### Myelomonocytic cells

BRAQUE$^{subclass}$ analysis yielded 15 myelomonocytic cell types (Figs 12A and B, S7 Fig.A), each one with an unique spatial distribution.

Classical monocytes were diffusely scattered throughout the medullary cords and the interfollicular interstitium (S7 Fig.A) with a LYZ + PU1 + CD16$^{neg}$ predominant phenotype (Table 6). Neutrophils shared LYZ and PU1 positivity, and in addition were CD10 + , CD16+ and CD86 + .

CD16 was expressed on three types of myelomonocytic cells: alternative monocytes [76], CD16 + macrophages and a subpopulation of Lyve1 perivascular macrophages, all residing within endothelium-lined spaces [76,83].

Lyve1 + CD16$^{neg}$ perivascular macrophages shared a similar phenotype and location as the CD16 + counterparts.

CD11c, HLA-DR and the macrophage markers CD163 and CD68 epitomize the phenotype of two major groups of macrophages: macrophages and phagocytes, respectively [84].

**Table 5. Description, tissue location, phenotype, numerosity of the non-myeloid innate immune cell types identified in six whole LN.**

| Cell type | Description | Location (% of cells) | Markers frequency in cell-type defined clusters (underscored >70% of cells) | | | Markers ranked 1st-5th (bold: >30% of clusters) | n. cells | n. clusters | UBERON | Othogonal validation Refs. |
| | | | 70-100% | 50-69% | 30-49% | | | | | |
| --- | --- | --- | --- | --- | --- | --- | --- | --- | --- | --- |
| **DC** | | | | | | | | | CL:0000451 | |
| cDC1A | Conventional Dendritic Cell type 1, full phenotype | inter-stitial, para-cortex (20–50%) | CD74, CD141, CLEC9A, IDO, LYZ, TIM3 | BCL6, CD11c, HLADR, MYC, PU1 | CD4, CD31, CD86, IRF8, MYCN, RORC | **IRF8, PU1, TIM3** | 31,451 | 51 | CL:0002394 | [16,55] |
| cDC1B | Conventional Dendritic Cell type 1, partial phenotype | para-cortex, med-ullary cords (20–50%) | CD141 | CD4, CD86, CD303, CLEC9A, MYC, PRDM1 | BCL2, BCL6, CD5, CD7, CD11c, CD16, HLADR, IDO, MYCN, PDL1, TIM3 | CD11c, CD86, cMAF, HLADR, OX40 | 33,312 | 17 | CL:0002394 | [16,55] |
| cDC1 prolif | proliferating conventional Dendritic Cell type 1 | inter-stitial (≥70%); para-cortex (20–50%) | BCL6, CD74, CD141, CLEC9A, ID2, IRF8, Ki-67, LYZ, MYC, PU1, RORC, TIM3 | IDO | HLADR, MX1, PRDM1 | BCL6, IRF8, PU1, PDL1 | 1,484 | 5 | | [16] |
| cDC2A | Conventional Dendritic Cell type 2, CD207+CD5neg | para-cortex (50–70%); inter-stitial (20–50%) | CD1A, CD86, CD207, HLADR, IRF4 | BCL6, CD1c, CLEC9A, PU1, RORC | CD74, PDL1, S100B | **CD103, CD207, IRF4, PU1, S100B** | 17,958 | 32 | CL:0002399 | [16,56] |
| cDC2A CD5+ | conventional Dendritic Cell type 2, CD207+CD5+ | para-cortex (≥70%); | CD1A, CD4, CD5, CD86, CD207, CLEC9A | BCL2, CD1c, CD45, HLADR, PRDM1, PDL1, S100B | CD7, MYC, OX40 | **CD1A, CD207** | 34,731 | 50 | CL:0002399 | [16,55,56–64]. |
| cDC2B | Conventional Dendritic Cell type 2, CD207neg, CD5neg | inter-stitial, para-cortex, capsular (20–50%) | CD86, IRF4, PU1 | CD1c, HLADR | BCL6, PDL1, RORC, TIM3 | AXL, CD31, CD45, HLADR, IRF4, PU1, S100B | 15,921 | 23 | CL:0002399 | [16,55, 57–64]. |
| cDC2B CD5+ | Conventional Dendritic Cell type 2, CD207neg, CD5+ | para-cortex (≥70%); | BCL2, CD4, CD5, CD86, HLADR, PRDM1 | CD1c, CD7, CLEC9A, MYC, PDL1 | CD1A, CD45, IDO, IRF4, MYCN, S100B | CD86, cMAF, HLADR, IRF4, OX40, PU1 | 24,007 | 36 | CL:0002399 | [16,55,57–64]. |
| DCmPh | Conventional Dendritic Cell, mixed type 1 and 2 phenotype | para-cortex (50–70%); | TIM3 | CD1c, CD11c, CD45, CD86, CD141 | AXL, BCL6, CD4, CD74, CD207, cMAF, CLEC9A, CLEC10A, HLADR, LYZ, MYC, MYCN, OX40, PRDM1, PU1, PDL1, RORC | HLADR, IRF4, PU1, S100B | 42,247 | 47 | | [56,62,65] |
| ID2+Ki-67+ | ID2+Proliferating cells, non T, non B | inter-stitial (≥70%); | ID2, Ki-67, PRDM1 | MYC | AXL, BCL6, cMAF, IRF4, PU1 | BCL2, CD16, ID2, IRF4, Ki-67, PRDM1 | 19,943 | 13 | | |

*(Continued)*

**Table 5.** (Continued)

| Cell type | Description | Location (% of cells) | Markers frequency in cell-type defined clusters (underscored >70% of cells) | | | Markers ranked 1st-5th (bold: >30% of clusters) | n. cells | n. clusters | UBERON | Othogonal validation Refs. |
|---|---|---|---|---|---|---|---|---|---|---|
| | | | 70-100% | 50-69% | 30-49% | | | | | |
| pDC | Plasmacytoid Dendritic cells | interstitial, paracortex, medullary cords (20–50%) | CD303, IRF8, MX1, TCF4 | CD31, CD74 | CD68, CLEC10A, IRF4 | CD86, IDO, IRF8, TCF4 | 63,090 | 37 | CL:0001058 | [16,55] |
| ILC3 | Innate lymphoid cells type 3 | interstitial (50–70%); paracortex (20–50%) | PU1, RORC | CD11c, CD74, CLEC10A | AXL, CD1c, CD45, LYZ, MYC, TIM3 | CD1c, CD11c, IRF4, IRF8, PU1, RORC, S100B | 46,938 | 45 | | [66–68] |
| NK | Natural killer cells | paracortex (50–70%); | CD16, CD56 | CD7, CD31, CLEC10A, MYC | AXL, CD5, HLADR, IDO, MYCN, OX40, S100B | **CD56, CLEC10A, HLADR, OX40** | 8,821 | 19 | CL:0000623 | [16,69] |
| DC undefined | DC cells of uncertain phenotype | paracortex, medullary cords (20–50%) | | BCL2, CD5, MYC | AXL, CD4, CD7, CD45, CLEC9A, CLEC10A, HLADR, MYCN, PRDM1 | CD4, CD5, CD11c, CD86, CD303, HLADR, IDO, IRF4, IRF8, MYCN, S100B, TCF4 | 93,032 | 51 | CL:0000451 | |
| Average DC | Default assignment by BRAQUE | NA (≥70%); | AXL, BCL6, CD31, CD56, CD74, CLEC9A, CLEC10A, MYC, OX40, TIM3 | CD14, CD11c, CD45, CD68, CD86, CD141, CD163, CD207, cMAF, HLADR, ID2, IDO, IRF8, LYZ, MX1, MYCN, PRDM1, PU1, PDL1 | CD1A, CD1c, CD7, CD16, CD103, CD303, IRF4, TCF4 | **CD207, TIM3** | 57,998 | 6 | | |
| ID2+Ki-67+B cells | ID2+Proliferating B cells (from BRAQUE2 on DC) | follicular (50–70%); interstitial (20–50%) | BCL6, ID2, Ki-67, MX1 | CD45, CD303 | CD74, PRDM1 | BCL6, CD11c, ID2, Ki-67 | 5,023 | 8 | | |
| Macrophages | Tissue macrophages (from BRAQUE2 on DC) | medullary cords (50–70%); paracortex (20–50%) | | LYZ | CD11c, CD303, CLEC10A | CD11c, CD163, IRF8, PU1 | 40,100 | 47 | | |
| Monocytes | Monocytes (from BRAQUE2 on DC) | interstitial (50–70%); | | | | BCL6, CD5, CD7, CD11c, CD31, CD86, CD303, CLEC9A, CLEC10A, ID2, IRF8, LYZ, PU1 | 39,288 | 12 | CL:0000576 | [16] |

Table 5 legend: See legend to Table 2.

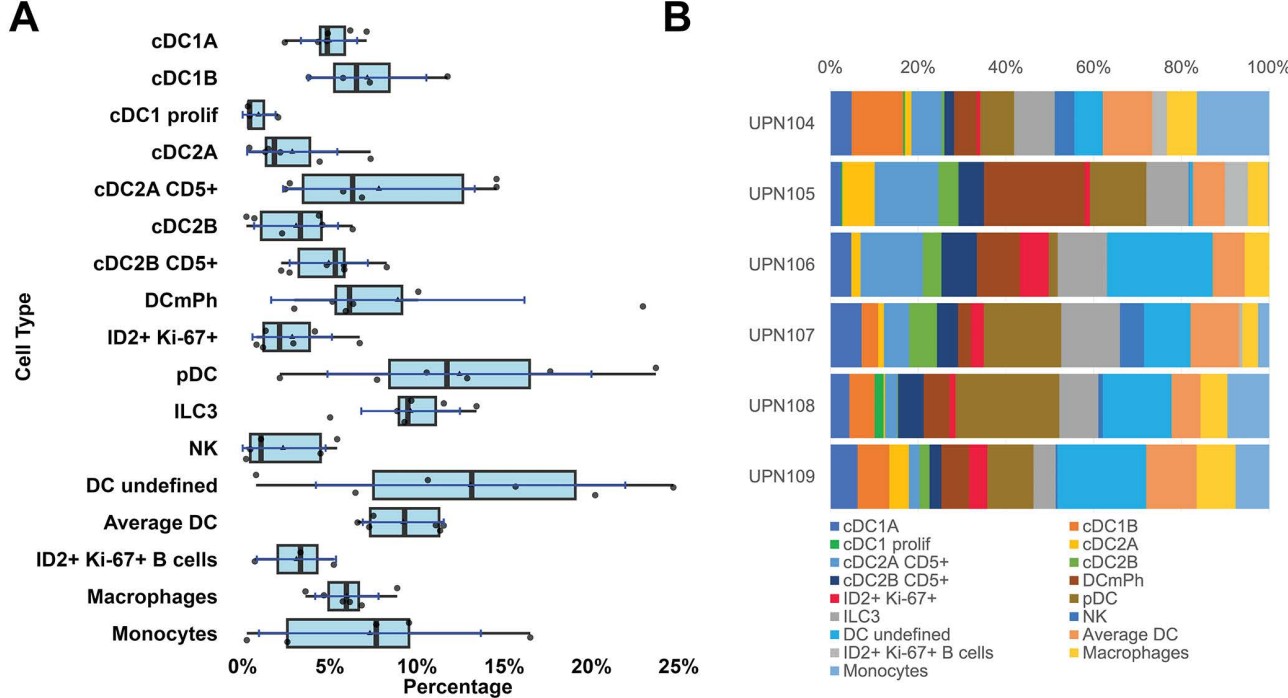

**Fig 9. Dendritic and innate immune cell classification. A**: Dendritic cell types from six whole LNs represented as boxplots. For the boxplots specifications see Fig 2. Average DC are identified by default. Abbreviations: cDC: conventional DC; DC mPh: DC with mixed phenotype; pDC: plasmacytoid DC; ILC3: innate lymphoid cells type 3; NK: natural killer cells. ID2 + Ki-67 + B-cells, macrophages and monocytes were misclassified as DC in the BRAQUE$^{global}$ analysis, but re-classified after BRAQUE$^{subclass}$. **B**: DC subtypes percentage composition of six whole lymph nodes.

Macrophages, the largest myelomonocytic cell type, had a preferential distribution in the medullary cords (S7 Fig.A), were more consistently represented in the LN examined and the phenotype was dominated by markers conventionally regarded as myelomonocyte-specific (CD14, CD31, CD68, CD163, TIM3, AXL) (Table 6).

Two types of cells had a consistent, highly expressed CD68 + phenotype: GC macrophages and interstitial phagocytes, often expressing CD86, MYC, PDL1, HLA-DR (Table 6) (S7 Fig.A).

BRAQUE found additional four cell types not previously described.

Two subsets had GZMB expression in common, one also displayed CD7, CD23 and IDO (Table 6), both were located in focal parenchymal aggregates (Fig 12C, S7 Fig.A), together with NK cells and where morphological signs of cell activation were noticeable (Fig 12D). The lineage affiliation of these two subsets is unknown.

Of the two other types, one (Act Macs) expressed MYC, TIM3 and other non-lineage specific biomarkers, and the other had a CD163, cMAF, PU1 phenotype (S7 Fig.A).

## Non-hematopoietic stromal cells identified in normal LN

Although non-hematopoietic stromal cells were not targeted specifically in the analysis, about a quarter of the markers evaluated were shared between lymphomyeloid and stromal cells (S1 Fig), a few were stroma-restricted, allowing a cell classification for the latter. CD248/Endosialin and vWF, both restricted to non-hematopoietic cells (megakaryocytes are

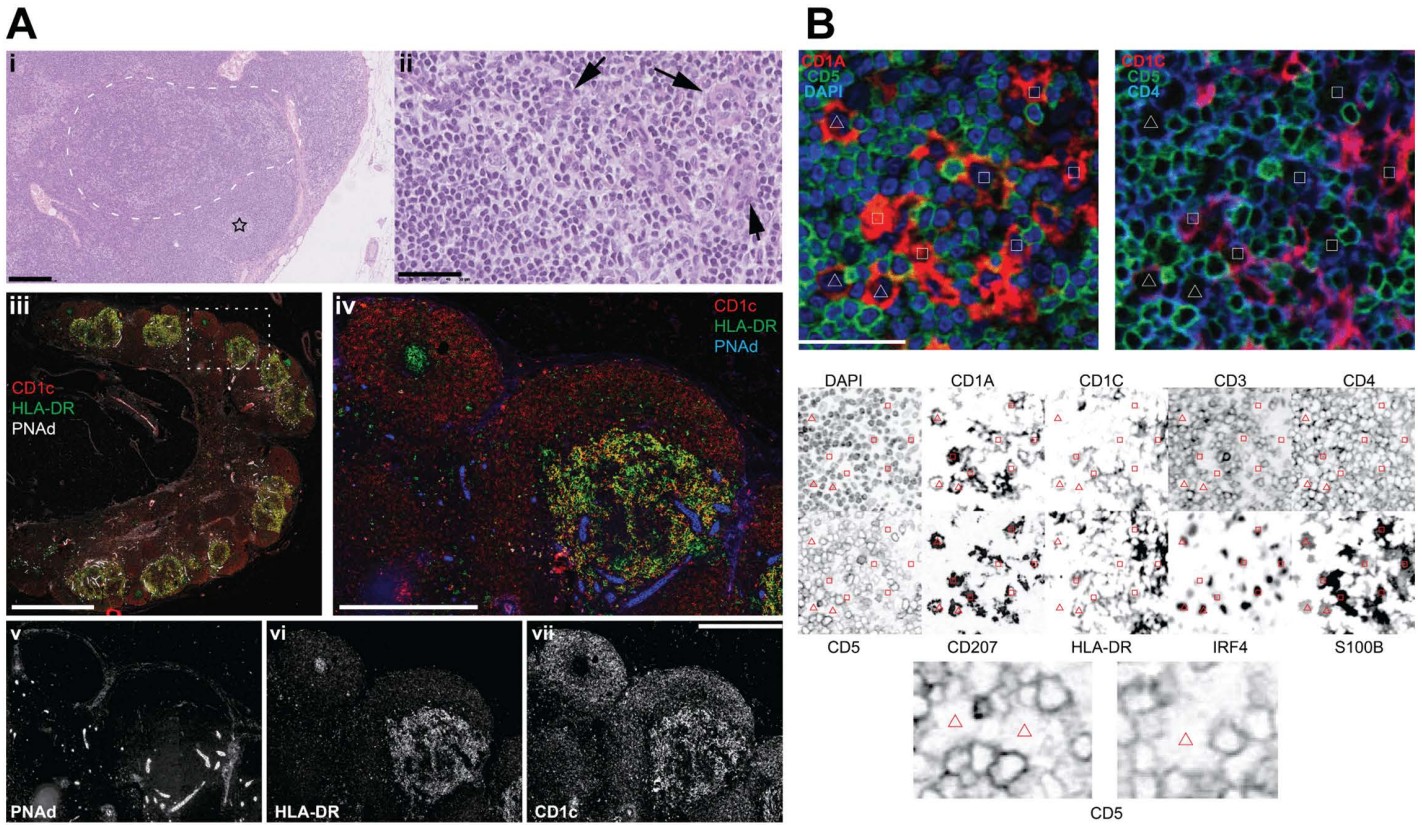

**Fig 10. Histopathology and immunophenotype in situ of fairy circles (nodular focal DC clusters). A:** Paracortical aggregation of cDC2 (Fairy Circles, FC). **i:** Low power H&E stain (UPN104). The dotted outline highlights a paler area in the paracortex. A star labels a small GC surrounded by a MZ. Scale bar: 250μm. **ii:** a high magnification of the dotted area in **i** shows a polymorphic cell population composed of small lymphocytes, larger blasts, larger cells with a typical indented nucleus and ample clear cytoplasm and some HEV (arrows). Scale bar: 50 μm. **iii:** low power image of LN stained in immunofluorescence for CD1c (red), HLA-DR (green) and PNAd (white). Scale bar: 250 μm. **iv:** the dotted rectangular area in **iii** is magnified. A FC in the lower right is composed of cells with a dendritic appearance and a variegation of CD1c (red) and HLA-DR (green) double expression (yellow). On the upper left, a brightly green (DR+) small GC is surrounded by a red CD1c+ follicular MZ. PNAd+HEV (blue) are surrounding and entering the FC. Scale bar: 500 μm. **v, vi, vii:** the individual immunofluorescence components of Fig **iii** and **iv** are shown individually as a grayscale. Scale bar: 500 μm. **B:** RGB and grayscale composite of a single cDC2-containing area. IF images for DAPI (blue), CD5 (green), CD1A (red), CD1c (red) and CD4 (blue) are composed as RGB color images (top half). The figure bottom half contains inverted grayscale IF images of the same area, stained for 10 relevant antigens.The polygons are reproduced in identical locations across all the images and identify cDC2. Three CD5-stained cells centered on the triangles are magnified. Scale bar 50 μm.

absent from LN), only labelled stromal and endothelial cells (Table 7 and S5 Table), providing proof of the single-cell specificity of both the segmentation and the BRAQUE pipeline.

Follicular dendritic cells (FDC), endothelial cells and fibroblasts, these latter named "stromal cells" because of absence of classifying markers (named NESC, non endothelial stromal cells by Abe et al.[9]) were subclassified into ten cell types (Table 7) (Figs 13A and B, S7 Fig.B).

Endothelial cells were classified according to the expression of CD31, shared with myeloid cells, and lineage-associated markers such as Lyve1 (this one restricted to lymphatic and sinus-lining endothelium), vWF, CD34, TCF4 and MYC. The last three were also found on non-endothelial stromal cells.

Lyve1neg capillary endothelium could be subdivided in two groups, high endothelial venules (HEV) and conventional endothelial cells because of tissue distribution and subtle phenotypic details. HEV were allocated in the paracortex, had

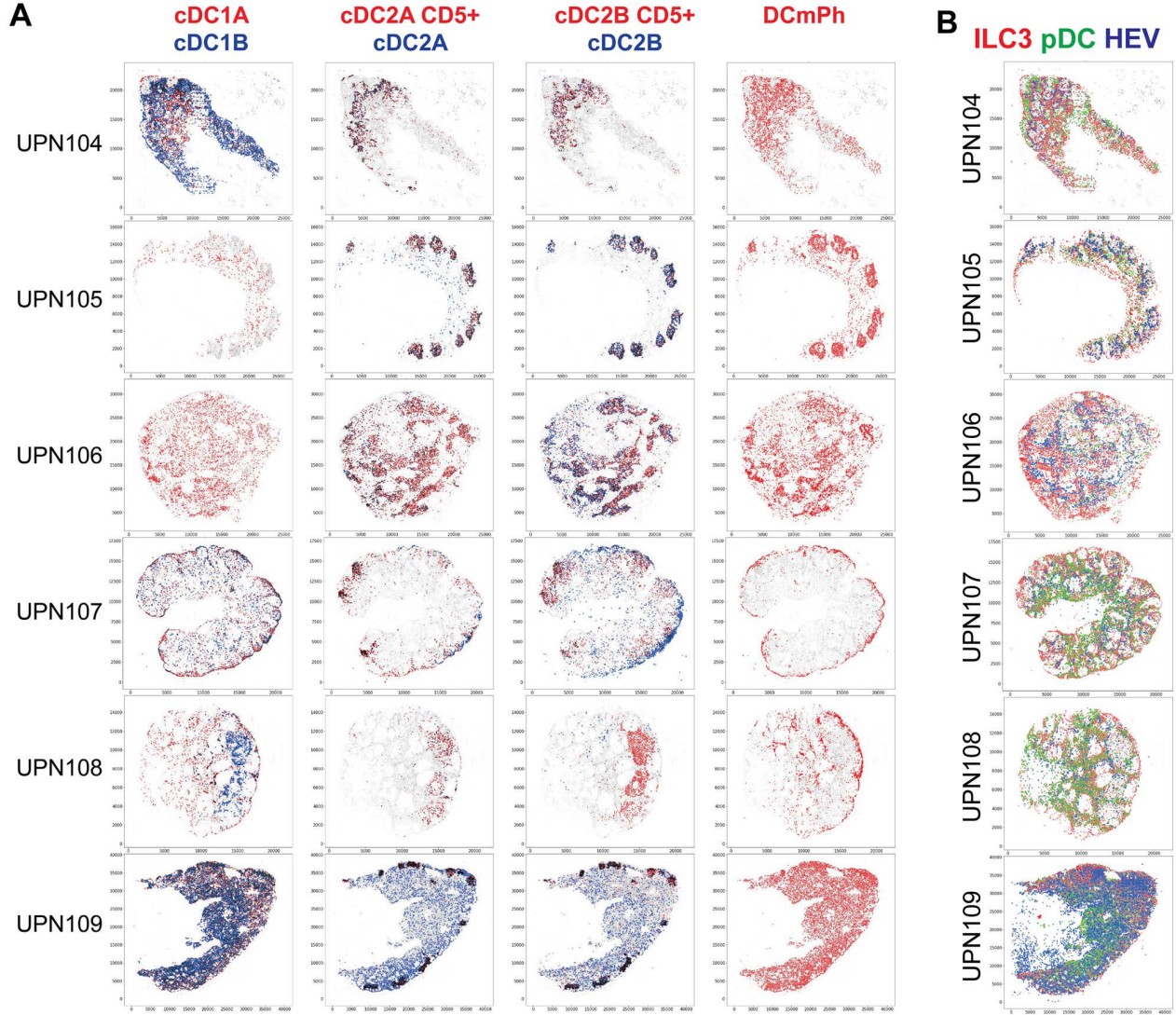

**Fig 11. Spatial allocation of DC cell types. A:** Seven DC subtypes are plotted, six of them as contrasting color-coded in pairs, spatially plotted. Cell types are color-coded. Cell overlap is in black. **B:** Spatial distribution of ILC3 and pDC. ILC3 (red), pDC (green) and HEV (blue) are spatially plotted on six whole LN. The scale represents pixels (0.45 μm/pixel).

shorter, broader branching, and expressed MYC. PNAd was invariably one of the first three ranking significant markers (Table 7). Of note, centrocytes were found adjacent to HEV (Fig 13C)

The coexpression of SOX9 and TCF4, together with CD248, AXL and AID defines LN fibroblasts, some MYC+, diffuse through the LN parenchyma or lining defined structural units (capsule, medullary sinuses, vasculature). The extralymphoid expression of AID has been documented in primary fibroblasts [89], albeit at 1/10th of the levels in B-cell lines. Coexpression of TCF4 and SOX9 message has been reported in fibroblasts, TCF4 only in endothelial cells (BioGps, http://biogps.org last accessed Feb 26, 2024) [86].

Some LNs contained a peri-follicular population of stromal cells with peculiar markers (CD11c, CD141, TIM3).

Extranodal stromal cells had a different phenotype, characterized by AXL, BCL6, cMAF and Zeb1.

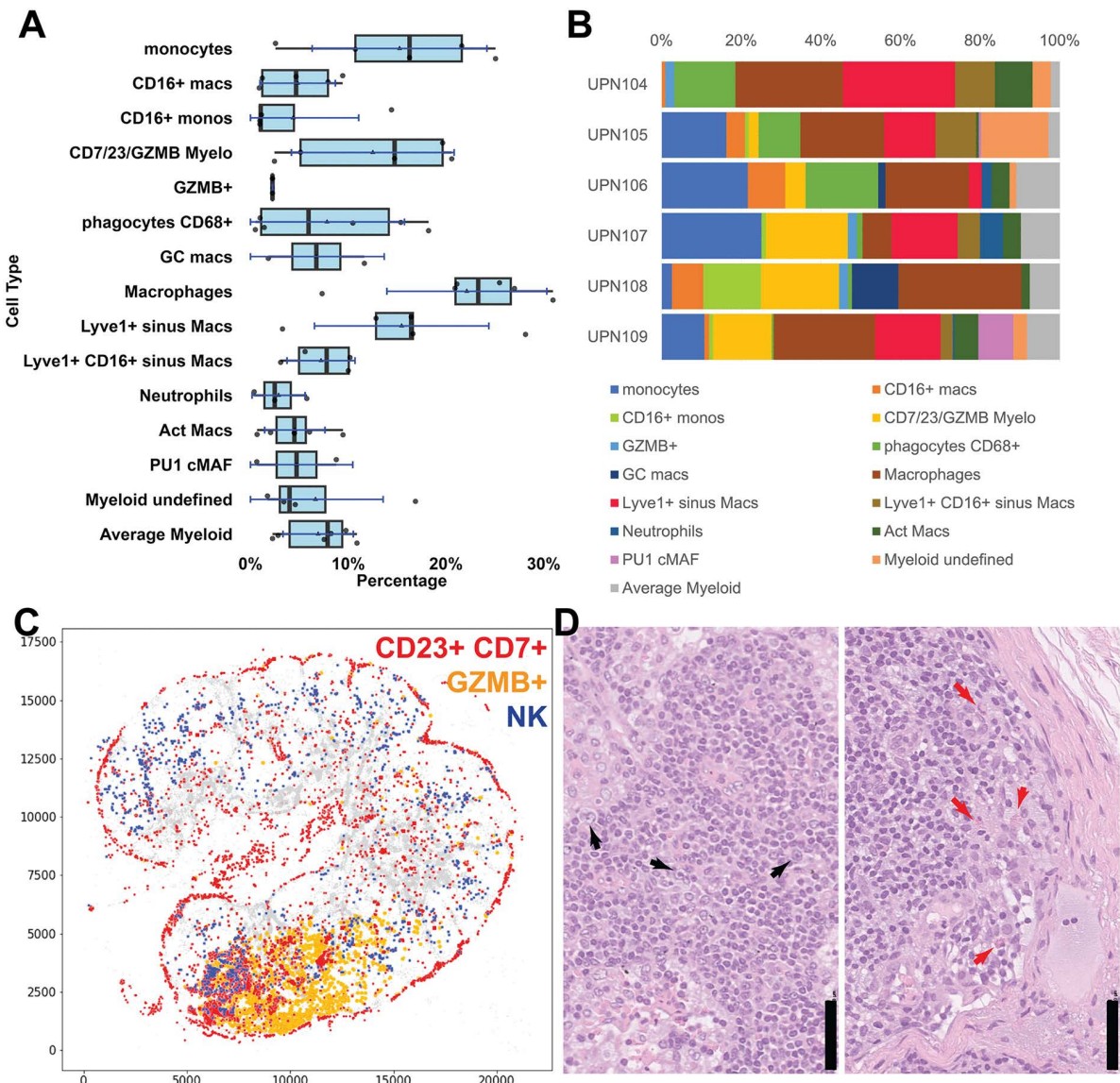

**Fig 12. Myeloid cell classification and spatial distribution. A:** Myeloid cell types from six whole LN, shown as boxplots. For the boxplots specifications see Fig 2. Average Myeloid cells are identified by default. Macs: macrophages; Act Macs: activated macrophages; PU1 cMAF: macrophages identified by high PU1 and cMAF. **B:** Myeloid cell subtypes percentage composition of six whole lymph nodes. **C:** The spatial distribution of CD23+CD7+GZMB+ (red), GZMB+ (yellow) and NK cells (blue) is shown on the outline of UPN107. **D:** High magnification of an H&E detail of UPN107 (left) shows haemorragic effusion, prominence of HEV and scattered large blasts (black arrows). Another H&E detail (right) shows a marginal sinus containing scattered eosinophils (red arrows). Scale bar = 50 μm.

## Neighborhoods of the normal human LN

We computed the statistical significance of the neighborhoods with two different approaches: firstly with a fisher test, significant when the neighboring cells were particularly enriched in a given cell type, and then with a method based on cellular "overlap", to better quantify stronger / weaker cellular spatial relations (Fig 14).

Together with statistical tests, we wanted to evaluate a further metric, the median distance to the nearest cell of and from each specific population (Fig 14, S9 Table and S10 Table), which may provide further information.

**Table 6. Description, tissue location, phenotype, numerosity of the myeloid cell types identified in six whole LN.**

| Cell type | Description | Location (% of cells) | Markers frequency in cell-type defined clusters (underscored >70% of cells) 70-100% | 50-69% | 30-49% | Markers ranked 1st-5th (bold: > 30% of clusters) | n. cells | n. clusters | UBERON | Othogonal validation Refs. |
|---|---|---|---|---|---|---|---|---|---|---|
| **Myeloid** | | | | | | | | | CL:0000763 | |
| monocytes | Monocytes | interstitial (50–70%); | LYZ, PU1 | CD14, CD45, CD86 | CD11c, CD74, MYC, TIM3 | AXL, BCL2, CD7, CD14, CD86, cMAF, LYZ | 95,970 | 30 | CL:0000576 | [16] |
| CD16 + macs | CD16 + macrophages | sinusoi-dal NOS (50–70%); | AXL, CD4, CD16, CD31, CD163, TIM3 | CD45, CD86, cMAF, MX1 | BCL2, CD11c, CD68, MYC, MYCN, PDL1 | AXL, CD16, CD163, LYZ | 16,979 | 15 | | [76,77] |
| CD16 + monos | CD16 + alternative monocytes | inter-stitial, medullary cords, sinusoi-dal NOS (20–50%) | CD16, CD31 | BCL2, CD45, MX1, MYC | AXL, CD11c, CD86, MYCN, TIM3 | AXL, BCL2, CD11c, CD14, CD16, CD45, CD103, LYZ | 9,730 | 6 | CL:0002397 | [76,77] |
| CD7/23/GZMB Myelo | Activated macrophages | Activation spot (≥70%); | CD7, CD23, GZMB, IDO | CD103, MX1 | AXL, CD14, CD74, CD86, MYC | CD7, CD16, CD23, CD68, GZMB | 86,575 | 44 | | [78–81] |
| GZMB+ | GZMB+ cells | Activation spot (50–70%); | GZMB | CD23, CD45, IRF8 | CD4, CD74, CD86, IDO, MYC | CD14, CD74, GZMB, HLADR, Lyve1, MX1 | 4,730 | 8 | | [78,79] |
| phagocytes CD68+ | CD68 + phagocyt-ing macrophages (also tingible body macrophages) | interstitial (50–70%); para-cortex (20–50%) | CD11c, CD68, CD74, CD86, MYC, PDL1 | CD4, CD45, HLADR, LYZ, PU1, TIM3 | AXL, BCL2, CD10, CD31, CD163, cMAF, ID2, MX1, MYCN | **CD68, CD163** | 30,714 | 20 | CL:0000888 | [16,77,82] |
| GC macs | GC macrophages | follicular (≥70%); | CD11c, CD45, CD68, CD86, HLADR, ID2, IRF8, PU1 | CD10, cMAF, Ki-67, MYC | CD23, CD74, PDL1 | **CD11c, CD68, CD86** | 5,189 | 6 | | |
| Macrophages | Resident tissue macrophages | medullary cords (50–70%); para-cortex (20–50%) | CD4, CD163 | BCL2, CD68, TIM3 | AXL, CD14, CD31, CD45 | CD163, cMAF, LYZ, PU1 | 140,103 | 29 | CL:0000235 | [77,82] |
| Lyve1 + sinus Macs | Lyve1 + perivascular macrophages | sinusoi-dal NOS (≥70%); | AXL, CD31, Lyve1 | CD163, cMAF, MYC | CD4, CD11c, CD14, MX1, MYCN, TIM3 | CD14, CD163, cMAF, Lyve1, LYZ | 96,071 | 46 | CL:0000887 | [77] |

*(Continued)*

| Cell type | Description | Location (% of cells) | Markers frequency in cell-type defined clusters (underscored >70% of cells) | | | Markers ranked 1st-5th (bold: >30% of clusters) | n. cells | n. clusters | UBERON | Othogonal validation Refs. |
|---|---|---|---|---|---|---|---|---|---|---|
| | | | 70-100% | 50-69% | 30-49% | | | | | |
| Lyve1 + CD16 + sinus Macs | CD16 + Lyve1 + peri-vascular macrophages | sinusoidal NOS (≥70%); | AXL, CD4, CD11c, CD16, CD31, CD163, Lyve1, MX1 | MYC, MYCN, TIM3 | CD14, CD45, CD86, cMAF | CD16, CD23, Lyve1 | 27,410 | 15 | CL:0000887 | [77] |
| Neutrophils | Neutrophils | interstitial, para-cortex, medullary cords, sinusoidal NOS (20–50%) | CD10, CD16, CD86, LYZ, PU1 | CD45 | | CD10, CD14, CD16, LYZ | 11,116 | 7 | CL_0000775 | [16] |
| Act Macs | Activated macrophages | para-cortex, capsular (20–50%) | MYC, TIM3 | CD4, CD74, CD86, HLADR, IDO | BCL2, CD11c, CD23, CD31, CD45, CD103, CD163, GZMB, ID2, IRF8, MX1, MYCN, PDL1 | CD10, CD11c, CD16, cMAF, MX1, PDL1 | 33,633 | 15 | | [16] |
| PU1 cMAF | PU1 + macrophages | interstitial (50–70%); para-cortex (20–50%) | CD163, cMAF, PU1 | AXL, CD11c, CD45, CD68, CD74, CD86, HLADR, MYC, MYCN, TIM3 | BCL2, CD4, ID2 | **cMAF, PU1** | 27,984 | 10 | | |
| Myeloid undefined | Myeloid cells of undefined phenotype | interstitial, para-cortex (20–50%) | MYC | AXL, BCL2, CD31, CD74, CD103, cMAF, ID2, MX1, TIM3 | CD4, CD10, CD45, CD68, CD86, CD163, MYCN | CD10, CD11c, CD14, cMAF, IRF8, Lyve1, LYZ, MYCN, PDL1 | 23,879 | 14 | | |
| Average Myeloid | Default assignment by BRAQUE | NA (≥70%); | HLADR, TIM3 | AXL, CD7, CD10, CD11c, CD14, CD16, CD23, CD68, CD74, CD86, cMAF, GZMB, ID2, IDO, MX1, MYC, PDL1, PU1 | CD4, CD31, CD45, CD103, IRF8, Ki-67, Lyve1, LYZ, MYCN | CD163, LYZ | 53,097 | 6 | | |

Table 6 legend: See legend to Table 2.

**Table 7. Description, tissue location, phenotype, numerosity of the stromal cell types identified in six whole LN.**

| Cell type | Description | Location (% of cells) | Markers frequency in cell-type defined clusters (underscored >70% of cells) | | | Markers ranked 1st-5th (bold: >30% of clusters) | n. cells | n. clusters | UBERON | Othogonal validation Refs. |
|---|---|---|---|---|---|---|---|---|---|---|
| | | | 70-100% | 50-69% | 30-49% | | | | | |
| **Stromal Cells** | | | | | | | | | CL:0000057 | |
| endothelium | Endothelial cells | diffuse, interstitial and extranodal | CD14, CD31, CD34, CD64, SOX9, TCF4 | AID, AXL, CD141, CD248, MYC, PNAd, vWF, ZEB1 | CD137, CD207, CD23, CD56, CD68, CLEC10A, cMAF, CXCL13, GZMB, MX1, OX40, PD1, PDPN, TIM3 | CD16, CD3, CD34, CD4, HLADR, vWF | 167,539 | 74 | CL:0000071 | [85–88] |
| HEV | PNAd+ endothelial cells | paracortex; intranodal | CD248, CD31, CD34, CD64, PNAd | AID, CD14, CD141, MYC, PD1, SOX9, TCF4, vWF | AXL, CD137, CD23, CD303, CD56, CD86, CLEC10A, CLEC9A, CXCL13, GZMB, LYZ, MX1, MYCN, OX40, PDL1, PDPN, TIM3, TOX1, ZEB1 | CD3, CD31, CD34, CD4, Ki-67, PAX5, PNAd, PU1 | 54,324 | 27 | UBERON:8410038 | [88] |
| Lyve1+endothelium | Lyve1+lymphatic endothelium | medullary cords sinusoidal lining and extranodal | AXL, CD163, CD303, CD31, CD64, cMAF, Lyve1, ZEB1 | AID, CD14, CXCL13, LYZ, TCF4 | CD137, CD4, CD68, lambda, MX1, MYC, MYCN, SOX9, TIM3, vWF | CLEC9A, Lyve1, TCRd, vWF | 59,949 | 26 | CL:0002138 | [88] |
| stroma (extranodal) | non-endothelial stromal cells in the perinodal space | extranodal | BCL6, cMAF, ZEB1 | AXL, CD248, CD34, TCRd | CD10, CD207, CD303, CD56, MYC, PDPN, S100B, TCF4 | CD16, FOXP3, IRF8, MYCN, PU1, S100B, ZEB1 | 48,707 | 50 | | |
| perifollicular stroma | non-endothelial stromal cells delimiting the B cell follicles | paracortical crowding around the B cell follicles | CD11c, CD14, CD141, TIM3 | CXCL13, MYC, ZEB1 | CD1c, CD248, CD303, CD31, CD64, CD74, CD79a, CLEC9A, cMAF, IgD, OX40, PAX5, PNAd, PDPN, SOX9 | CD14, CD141, CD206, CD31, CD79a, PNAd | 70,458 | 11 | | |
| stromal cells | non-endothelial stromal cells (fibroblasts, myofibroblasts, FRC, smooth muscle cells) | diffuse, interstitial | AID, AXL, CD248, CD34, CD5, CD56, CD64, CLEC10A, MYC, PNAd, SOX9, TCF4 | CD103, CD137, CD21, CD23, CD31, CD74, cMAF, GZMB, MX1, PD1, TOX1 | CD11c, CD14, CD303, CD68, CD7, EOMES, IDO, LAG3, Lyve1, PDPN, TIM3, ZEB1 | **AXL** | 277,535 | 100 | | [16,89] |

*(Continued)*

**Table 7.** (Continued)

| Cell type | Description | Loca-tion (% of cells) | Markers frequency in cell-type defined clusters (underscored >70% of cells) | | | Markers ranked 1st-5th (bold: >30% of clusters) | n. cells | n. clusters | UBERON | Othogonal validation Refs. |
|---|---|---|---|---|---|---|---|---|---|---|
| | | | 70-100% | 50-69% | 30-49% | | | | | |
| stromal cell MYC+ | MYC+non-endothelial stromal cells (fibroblasts, myofibroblasts, FRC, smooth muscle cells) | diffuse, intersti-tial | AID, AXL, CD137, CD14, CD207, CD23, CD248, CD31, CD34, CD56, CD64, CLEC10A, cMAF, CXCL13, GZMB, MYC, OX40, PD1, PNAd, PDPN, SOX9, TCF4, ZEB1 | CD10, CD11c, CD141, CD69, TIM3, TOX1 | CD103, CD1c, CD21, CD7, LYZ, MX1, PDL1, PRDM1 | CD206, CD248, vWF | 37,622 | 25 | | [89] |
| FDC subpop | | | BCL6, CD21, CD23, CD4543, CD74, CD79a, CD86, CXCL13, PAX5, PD1, PDPN, TOX1 | CD20, GZMB | AID, CLEC10A, IgD, LAG3, MX1, MYC, MYCN, PRDM1, PU1, SOX9 | **CD21, CD23, CD248, PNAd** | 4,071 | 5 | CL:0000442 | [16] |

Table 7 legend: See legend to Table 2. Note that these data have been obtained after BRAQUE$^{global}$ analysis.

We could confirm known neighborhood relationships such as CD4 T-cell subsets (S100B+ effectors and activated Tregs) with cDC2 [90–92], centrocytes and MZ B-cells with DC [93,94], plasma cells with macrophages [95], the heteroaggregation of light chain positive plasma cells, and the self-aggregation of several, but not every cell type [96].

Other described neighborhoods were not confirmed: cDC1 and CD8 [24], cDC1 and HEV [70], macrophages and fibroblasts [97], ILC3 and endothelium or plasma cells [98]. Several CD4 and some CD8 T-cell types bound to endothelium, but none to HEV [99].

The reason for this may be a combination of LN selection, the strict significance criteria and/or the insufficient number of samples for a specific immune challenge.

Unanticipated novel neighborhoods were found.

A previously undescribed TCF7$^{lo}$ CD8 T-cells co-localized with CD8 showing IFN response (Fig 5).

IgD+CD27$^{neg}$ MZ B-cells are neighbors of plasma cells (Fig 6C, S5 Fig.B).

Centrocytes interacted with HEV and endothelium, the normal counterpart of the morphologic detail (penetration of GC by vessels) seen in unicentric hyaline-vascular variant of Castleman's disease [100] (Fig 13C)(S8 Table).

## The landscape of the normal human LN

By computing the overlap of each cell type versus each other, we draw a detailed landscape of the LN immune architecture (Figs 5 A and B, Figs 6 C and D, Fig 14, S8-S10 Figs.). In general, each cell type overlaps with 4.2±3.3 (range 0–18) of the other cell types, pointing to a very specific landscape occupancy. This is very informative of both the shared and the mutually exclusive spaces.

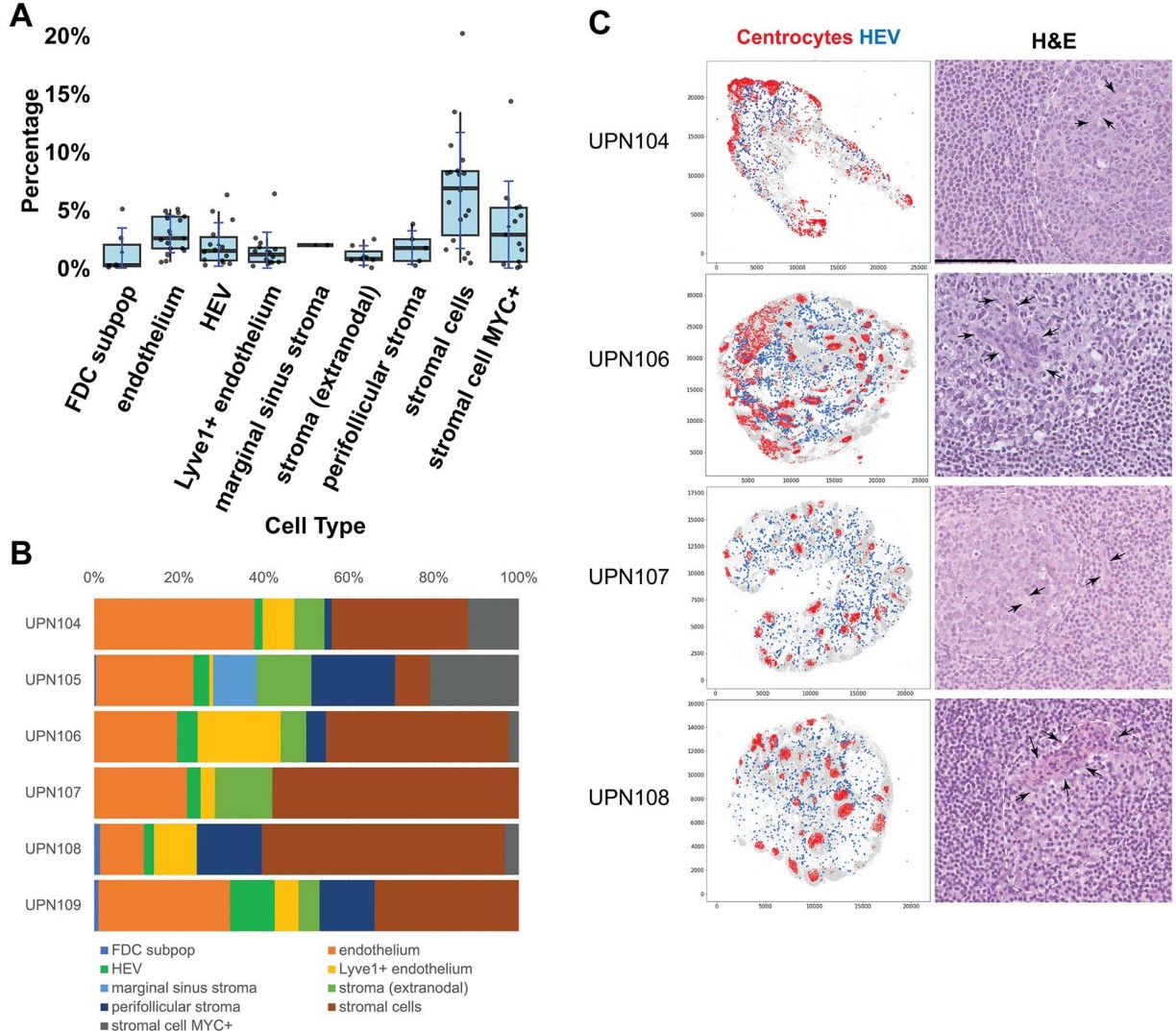

**Fig 13. Stromal cell classification and centrocyte-HEV spatial relationship. A:** Stromal cell types from six whole LN shown as frequency ±SD obtained by BRAQUE<sup>global</sup> analysis. For the boxplots specifications see Fig 2. **B:** Stromal subtypes percentage composition of six whole lymph nodes. **C:** Spatial distribution of HEV (blue dots) and centrocytes (red dots) in four whole LN (left column). Scales are pixels (0.45 μm/pixel). An H&E magnified representative field is shown in the right column. The dotted white line highlights a germinal center, the arrows point to HEV. Scale bar 100 μm.

An analysis of the statistically significant distribution of the cell types in the LN zones (Fig 15) reveals a few specific positive locations and several specific avoidances, not entirely unexpected for *a)* motile cells *b)* in LNs whose immune status is heterogeneous.

Non-naïve CD4 TCF7$^{hi}$ or TCF7$^{average}$ T-cells occupy the LN paracortex, a landscape in which no other cell types were preferentially located, except cDC1 and cDC2. Exhausted CD4 spread to the medullary cords and activation spots and did not overlap with other cell types (Fig 5A and B).

CD4 T$_{fh}$ cells, centroblast, centrocytes and the GC-related B-cell subsets occupy a GC environment, also shared with plasmablasts and GC macrophages.

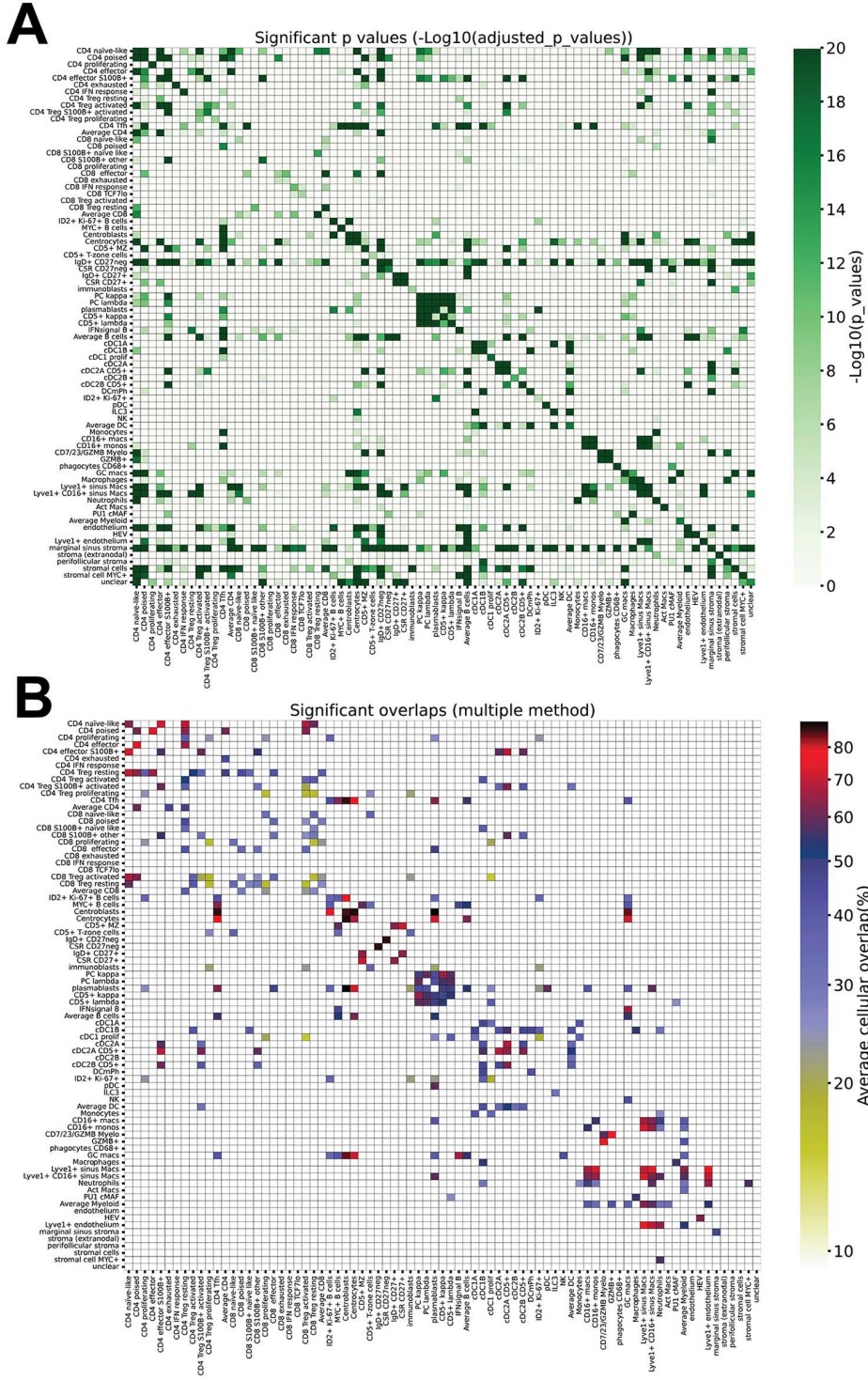

**Fig 14. Neighborhood and cell type overlap. A:** Neighborhood relationship between cell types. The figure contains all statistical significance values for each cell type (rows) versus the others (columns). Only the values equal or smaller than p 8.094544277157197e-06 (approx. 0.0000081) are shown. Color scale at the right. **B:** Overlap for each cell type (rows) versus the others (columns). Only the values equal or smaller than p 8.094544277157197e-06 (approx. 0.0000081) are shown. Color scale at the right.

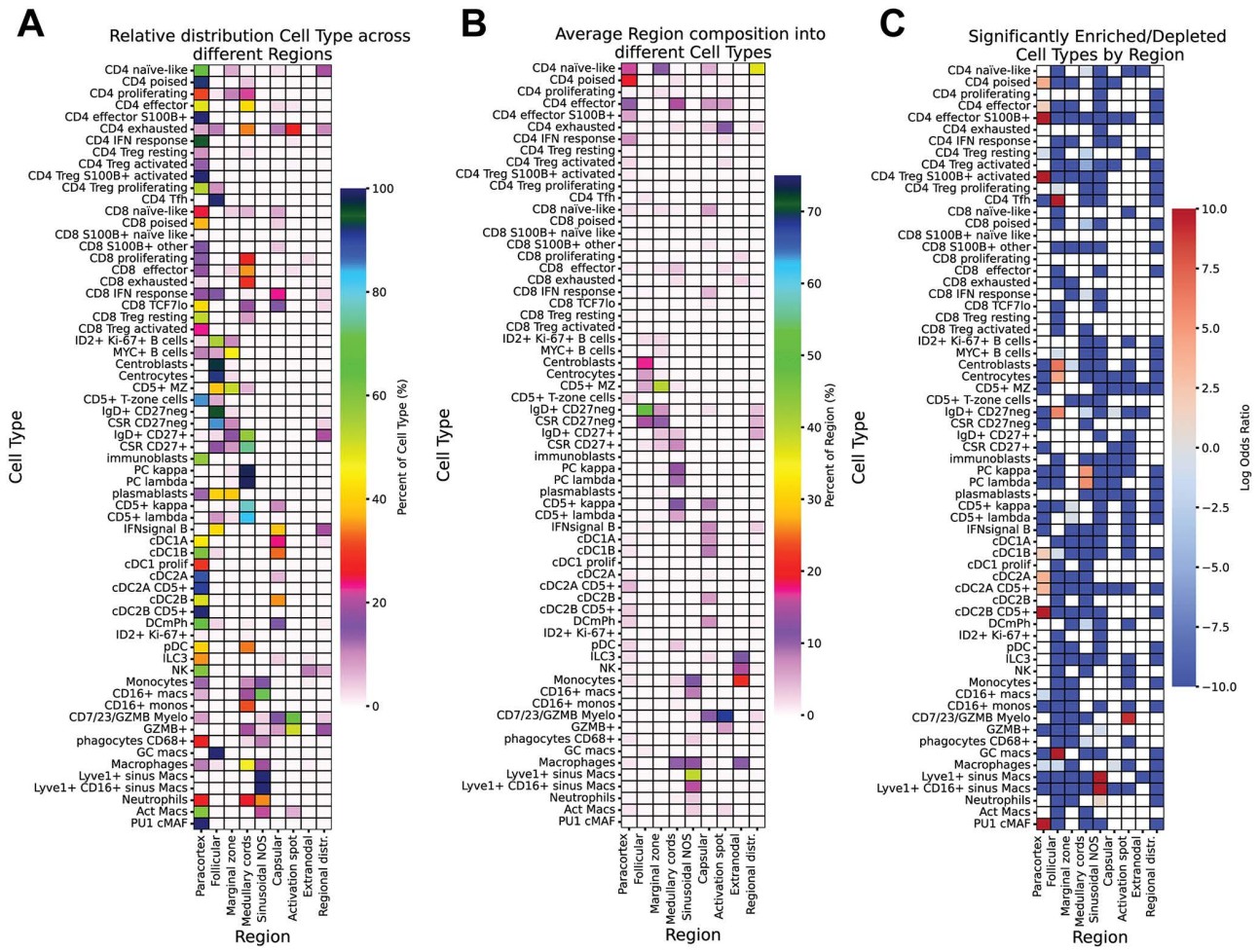

**Fig 15. Distribution and statistical significance of cell types across the LN regions. A**: Relative distribution of each cell type across different regions. Each horizontal line sums up to 100%. Scale on the right. **B**: Average region composition in terms of cell types. Each vertical column sums up to 100%. Scale on the right. **C**: Significantly enriched/depleted cell types per region, showing average log odds ratios from Bonferroni corrected fisher tests. Scale on the right. Data for UPN104–108.

The analysis of mature B-cell overlap revealed five non-overlapping spaces (Figs 6 C and D).

The first space comprises a dual aspect: GC-based centroblasts and centrocytes, but in addition largely extrafollicular ID2 + Ki-67+ and MYC+ cells admixed to centroblasts, a structure which may consist in the outer zone (Fig 7B) [101].

The second space contains CD5 + CD27neg putative T1 B-cells, but also MYC + B-cells (Fig 7B and C). A "marginal zone" definition may fit the distribution of CD5 + MZ B-cells.

The third hosts CD27neg IgD+ or negative B-cells and corresponds to the MZ (Fig 7A).

The fourth space is occupied by CD27 + MBC and may represent a newly described memory B-cell zone (MBZ) (S5 Fig.B). This zone was not included among the established LN zones, thus will not appear in Fig 15 and S12 Table.

The last corresponds to plasma cells, plasmablasts and the CD5 + light chain+ B-cells in the medullary cords.

cDC2A and CD5 + cDC2B share a space with each other, S100B + CD4 effectors, S100B + CD8$_{others}$ and CD4 T$_{reg}$s.

These cDC2 cells in real space revealed circular paracortical structures with an empty center, reminiscent of the nodules described before [102] (Fig 10A). We expanded the morphological analysis of these structures to 12 whole LN sections and 99 cases of various non-neoplastic LN pathology or morphologic variations of normalcy, by using robust cDC2 markers (CD1c and HLA-DR) and reviewing at low power the IF images (S13 Table). Because of the morphological reminiscence of these circles with another biological phenomenon (https://en.wikipedia.org/wiki/Fairy_ring), we named these "Fairy Circles" (or Focal Clusters; FC).

Well organized FC were visible in 92% of 12 whole sections and 56% of 16 2 mm TMA cores from "normal" LN. FC were absent from 75% of pathological TMA cores and from all the pathological whole sections images (S13 Table). Individual cells from these latter samples were not analyzed, therefore we cannot comment whether the disappearance of FC is mirrored by qualitative/quantitative changes in cDC2.

As described previously [102,103], PNAd+ High Endothelial Venules (HEV) were found close to or encroaching the FC (Fig 10A and S11 Fig.).

The perivascular space is occupied by CD16 +, Lyve1 + perivascular macrophages, the latter sharing the space with Lyve1 + endothelium.

GZMB+ cells tend to occupy a space by themselves, shared with PU1 + Macs.

## Discussion

We have used a 78 antibody-strong panel, an epitope- and tissue-saving technique, MILAN [12,104], and a recently developed novel bioinformatic pipeline, BRAQUE [10], to classify the cells constituting the human LN into 77 cell types according to known phenotypes and published and unpublished spatial landmarks. The integration of the phenotype and the tissue distribution of the individual clusters from BRAQUE has been fundamental for an extended cell identification.

The classification obtained features a combination of surface, cytoplasmic and nuclear proteins detected at high sensitivity [105], thus providing an invaluable proteomic granular reference dataset for molecular or multi omic data integration.

This is the most detailed classification of the whole LN performed in situ up to date and represents LN "normalcy" by size, histopathology and clinicopathologic standards [106]. Size has been the only parameter useful to classify a LN as "normal" [106,107]; we can now add preserved immune structures (FC) or phenotypic traits (absence of GZMB from pDC) as hallmarks of normalcy.

The phenotypes and the cell types we describe have been extensively cross-validated in years of immunology research, quoted throughout the text/tables and, more recently, by scRNAseq studies of the SLO within the HCA initiative [16].

Interestingly enough, this latter comprehensive study, which orthogonally validates the vast majority of our cell types, fails to identify IFN signal in T cells, transitional and B1 B cells, alternative macrophages and other cell types probably because of a weak correspondence between RNA and proteins [108,109] and more favorable representation of biology by proteins, compared to RNAs[109]. The fact that transcriptomics may be less suited for the classification of lymphoid tissue is confirmed by recent studies [34,110].

In the Caron study [34], a 265 antibody CITEseq panel aimed at surface markers shares with us 44 antibodies only, and lacks transcription factors and intracellular markers: it fails to identify in peripheral blood several lymphoid cell types we describe in the LN. While transcription factors alone do not suffice to classify cells (our data) other factors may be the cause besides the lack of intracellular markers: an elaborate signal thresholding algorithm (vs no gating in BRAQUE), and the focus on peripheral blood (vs LN).

We provide spatial relationship data and anatomical allocation for numerous cell types for which, in the authors' knowledge, only transcriptomic studies on cell suspensions or some single staining information were available. This is the case

for putative B1 or T1 cells, memory B-cells, CD5 + cDC2 and several subtypes of T-cells with function-associated pheno-types. In addition, we suggest a dedicated nodal site for memory B-cells, which we named memory B-cell zone (MBZ).

A nodal equivalent to the splenic marginal zone has been described in the subcapsular and medullary LN sinus in mice [111] and in parasite-infected humans [112]; these reports hint at a subcapsular sinus location for MBC in humans. We found clusters with such spatial distribution in 2 LNs only.

Between 1986 and 2001 hematopathologists [113–115] have described in non-pathological LN an indistinct and vari-able collection of cells with clear cytoplasm (monocytoid) located outside the mantle zone and bearing phenotypic features similar to the splenic marginal zone. This may be an earlier description of a nodal location for MBC, distinct from the MZ, based however on a very limited phenotype.

DN B cells (IgD$^{neg}$ CD27$^{neg}$) have elicited considerable attention lately: they have been linked to autoimmune disorder, chronic infections (HIV, malaria) and cancer [45,43] and thought to belong to an extrafollicular antigen response. To co-locate DN B cells with naïve MZ B cells in the follicle, apart from MBZ cells, is a novel finding.

The elusive CD5 + B1-like human B cell population is resolved in spatially and phenotypically distinct cell types, expressing TCF7, a finding very recently confirmed in blood and peritoneal effusions, but not in SLOs [116].

A combination of immunophenotyping, high-dimensional analysis and traditional histopathology has detailed circular gathering of cDC2 reminiscent of the "fairy circles".

These novel descriptions should prompt additional research to elucidate the composition, the stromal underlying and the dynamic of these novel zones.

A hyperplexed targeted Ab-based in situ proteomic investigation inherently relies on Ab specificity [117] and sensitivity, posing novel challenges [105]. The profile of each cell type in this study is obtained by listing the most frequently reported markers among the numerous clusters belonging to that type, not the most highly expressed. In BRAQUE these latter correspond to higher "effect size" [10,105] and are reported in a separate column in the Tables. Keeping that in mind, we faithfully reproduced established phenotypes, thanks to the redundancy of the panel used, but also listed are markers which have been reported only in experimental condition or in cell suspen-sions and never in FFPE tissue.

That is the case for CD23 in T-cells and myelomonocytic cells [78,118], CD7 on myeloid cells [81], TCF4 on endothe-lium [119], AID on fibroblasts [89] and CD5 on B cells and DC.

We report coexpression of markers (e.g., BCL6 and PRDM1) initially described to be mutually exclusive [31,48,120], but later revised in some cell types [121,122]. The protein half-life [108,109,123], the mutually inverse gradients of expres-sion [123–125] and the aggregation of multiple clusters in a single cell type may substantiate the results obtained by BRAQUE. Lastly, BRAQUE scores as statistically relevant values not appreciable by signal thresholding and/or visual assessment [105].

Canonical markers are conspicuously missing in BRAQUE$^{subclass}$: CD3, CD5, CD7 and BCL2 in T-cells, IRF4 on MZ B cells [48], CD30 on B-cells [126], etc. This is because BRAQUE lists the statistically significant markers which make each cluster unique and not the antigens broadly expressed in the target population, thus not making the statistical threshold [10,105]. Therefore, the phenotypes defining cell types in this work need to be integrated with canonical markers for tasks such as single cell manipulation.

## Limitations of the study

The neighborhood analysis of this study represents a "minimalist" view of the interactions, given the very strict require-ments to produce a significant score (10 AND 20μm, significant in all LN examined); tissues harmonized for a specific immune response may disclose additional meaningful interactions.

For some phenotypes a definite explanation is not available and deserves further study. This is the case, e.g., for CD23, which ranks among the five top-ranking markers in FDC and CD27neg CSR, where it is expected, but is listed on

CD27 + MBC, where it should not be according to previously published data [41]. The possible reasons may be Ab specificity, the inner working of the dimensionality reduction algorithm or the post-transcriptional regulation of CD23, which is particularly complex [127] in tissues. CD23 + Follicular dendritic cell embrace of CD23$^{neg}$ B cells may also explain these findings.

One may also consider that secreted/soluble proteins do diffuse in tissues (e.g., GZMB [128], vWF [129]) and are detected by BRAQUE, independently from the segmentation inaccuracies.

Other limitations of this study encompass the failure to identify cell types minimally represented in quiescent LN such as $\gamma \delta$ T-cells, double negative T-cells, ROR $\gamma \tau$ + T-cells, AXL + DC [130] and NK cell subsets [131]. The reasons may be the insufficiently diversified markers panel, the unappreciated broader distribution of markers described to be dichotomic in expression, the limitation of the segmentation protocol, the inherited problems of fixed and embedded material, unknown limitations of BRAQUE or a combination of all of that.

This study will help to consolidate an holistic view of an "average" human LN landscape, if ever such a definition may apply to the immune system in general, allowing the comparison with immune perturbations, particularly the ones not driven by clonal expansions, but fixed in timing and in stimulus drivers.

## Materials and methods

### Human specimens

Lymph node excisional biopsies and clinical notes were extracted from the laboratory information systems of the collaborating centers by the Authors with clinical privileges and anonymized. The inclusion selection criteria were: *i*) absence of pathology, inflammatory, autoimmune or neoplastic, *ii*) non-specific nodal changes in the histology description (e.g., paracortical or follicular hyperplasia, sinus histiocytosis), *iii*) any anatomical location, *iv*) uninvolved sentinel LN from breast or lower GU tract, or incidental finding, *v*) small size; the chosen samples measured an average = 0.98 cm, SD +/- 0.35 cm for the shorter axis, n = 19; for six whole LNs: average = 0.87 cm ± 0.28, range: 0.62–1.41 for the shorter axis (S1 Table). Note that the size reference values for normal LN does not exceed 1 cm. for the shorter axis [106,107], irrespectively of the organ site [107].

A progressive UPN (Unique Patient Number) was assigned to each patient. Paraffin blocks and sections to be analyzed were selected by a Pathologist after a review of the Hematoxylin and Eosin (H&E) stain. Only archival formalin-fixed, paraffin embedded material (FFPE) was used.

The clinicopathologic and biometric data of the cases comprising the dataset are listed in S1 Table.

None of the LN contained particulate material [132].

The study has been approved by the Institutional Review Board Comitato Etico Brianza, N. 3204, "High-dimensional single cell classification of pathology (HDSSCP)", October 2019. Consent was obtained from patients who could be contacted or waived according to article 89 of the EU general data protection regulation 2016/679 (GDPR) and decree N. 515, 12/19/2018 of the Italian Privacy Authority. Data and sample retrospective collection started after November 21st, 2019, after de-identification of protected data.

### Tissue Microarray (TMA) preparation

Tissue microarrays (TMA) were prepared as previously published on a Tissue Microarrayer Galileo model TMA CK4600 (RRID:SCR_024393) (Integrated System Engineering S.r.l., Milan, Italy). Cores of 2 mm were used, from selected areas annotated by a Pathologist. Duplicate cores were placed in two separate TMA recipient blocks.

### Histology repository

Virtual images of H&E stained sections, serial to immunostained ones, were deposited in a NDPIserve (RRID:SCR_017105) (Hamamatsu Photonics), available at: https://tiny.cc/LNproject

## Anatomical terms and definitions

The anatomical boundaries or the LN were defined as previously described [1,133]. The terms used throughout the manuscript are detailed in S14 Table. A UBERON terminology (https://evs.nci.nih.gov/ftp1/UBERON/About.html; last consulted on September 4th 2024) [134] was added when suitable (Table 1–7-7, S1 and S14 Tables).

## Antigen retrieval

Antigen retrieval (AR) was performed once placing the dewaxed, rehydrated sections [12] in a 800 ml glass container filled with the retrieval solutions (EDTA pH 8; 1 mM EDTA in 10 mM Tris-buffer pH 8, Merck Life Science S.r.l.,Milano, Italy; cat. T9285). The treatment, according to established protocols [135,136], consists in heating in a household microwave oven at full speed for 8 min, followed by intermittent electromagnetic radiation to maintain constant boiling for 30 min, and cooling the sections to about 50° C before use.

## Antibody validation and staining

Antibodies were validated according to published protocols [117,137]. A list of antibodies is reported in S2 Table, which also contains the cycle round number and the fluorochrome for the secondary antibody.

Representative low-power and high-magnification images of a whole LN staining is provided in S1 Fig.A, B.

The full panel consisted of 97 immunostains (+ EBER and autofluorescence). 20 antibodies who produced inconsistent staining were excluded from the analysis (S6 Table). Viral biomarkers (EBV, HHV8) were not analyzed in this dataset.

As an additional tool to explore expected relationships among variables (antibodies), a correlation matrix was constructed for sample UPN 107 .csv as a reference.

The R function "corplot" (order = AOE) was used for the representation. It provides a coloured table which represents the positive or negative correlation of one single marker versus the others. The order of markers is established according to the AOE parameter (the angular order of the eigenvectors) which is supposed to group markers with the same behavior.

A coloured side bar (representing the IF channel of acquisition of each marker, green = FITC, yellow = TRITC, RED = Cy5, BLUE = BV480) with a number (the staining round) is reported next to the biomarker (S1 Fig.C).

The preferential cell type distribution of all antibodies is reported in Supplementary S1 Fig.D.

## MILAN Immunofluorescence

Multiple immunofluorescent (IF) labeling was previously described in detail with the MILAN (Multiple Iterative Labeling by Antibody Neodeposition) method [12,104]. A detailed method has been published (https://dx.doi.org/10.21203/rs.2.1646/v5).

Briefly, the sections were incubated overnight with optimally diluted (1 µg/ml) primary antibodies in species or isotype mismatched combinations (e.g., rabbit + mouse, mouse IgG1 + mouse IgG2a, etc.), washed and counterstained with specific distinct fluorochrome-tagged secondary antibodies [12]. The list of primary and secondary antibodies is in S2 Table. The slides, counterstained with DAPI and mounted, were scanned on an S60 Hamamatsu scanner (Nikon, Campi Bisenzio, FI, Italy) (RRID:SCR_022537) at 20x magnification (0.45 µm/pixel) 8 bit. The filter setup for six color acquisition (DAPI, BV480, FITC, TRITC, Cy5) plus autofluorescence (AF) was as published [138]. After a successful image acquisition, the sections were stripped according to the MILAN method [104] and stained with another round.

Five slides (either whole sections of TMA) were processed simultaneously as described [104] to minimize intersample variations. Staining variability for the same Ab set after ~30 cycles was less than 15% and no method-dependent cell loss was noticed (see Supplementary Fig. 1 in Manzoni et al.[139]).

## Preparation of immunofluorescent images for single cell analysis

A pipeline, A.M.I.C.O.[140,141], was adapted and used to register all the images belonging to a case or a TMA and rename each image with the biomarker name, according to a signpost-containing file name and fluorescence channel saved at acquisition.

After the stainings were acquired, digital slide images (.ndpi) were imported as uncompressed .tiff with ImageJ (ImageJ, RRID:SCR_003070). Tissue autofluorescence (AF) was subtracted when appropriate as published [12]. In order to remove the contribution of circulating polyclonal immunoglobulins (versus the monoclonal membrane-bound or cytoplasmic Ig) [142], after AF subtraction, images for lambda light chains were subtracted from kappa light chains images, after equalizing the intensity of the former; the same was done for kappa light chains. The resulting image (kappa sine lambda and lambda sine kappa) were used for the analysis.

No other digital image modification was applied

## Validation of the analytical pipeline

**Cell segmentation.** A DAPI nuclear stain-based, Matlab-based segmentation algorithm, CyBorgh [10], was applied. CyBorgh algorithm output is a .csv file obtained by applying an expanded-nuclear mask to all registered images.tiff (one for each marker) belonging to a sample. In detail, for each cell, it is extracted the mean value among pixels (8 bit grayscale) related to each marker/image.

Data extracted for each single cell x image associated with spatial x and y coordinates composed the final matrix used for the analysis. Columns represent the variables (mean value per marker and spatial information), row the cells.

This algorithm was compared with Cellpose2 [143] (https://www.cellpose.org/; latest access 25 december 2023) by segmenting three 2 mm. TMA LN cores (UPN26, 32, 33) with both and comparing the cluster output and composition (S12 Fig.A and S15 Table).

The number of unrecognizable clusters, or spurious hybrid phenotypes (e.g., CD3 + CD20 + cells) and the failure to identify known subsets were used to rate each algorithm.

The code for CyBorgh can be downloaded from https://doi.org/10.17632/3ntbp3zdzh.2.

## Artifact identification and subtraction with BRAQUE

HDBSCAN, which is part of the BRAQUE pipeline [10], identifies outliers (e.g., cells having marker profiles heavily different from everything else, and similar to nothing else within the sample) as noise points and allocates them by default as a "-1" cluster, which acts as a garbage collector, and whose numerical dimension has been calculated for each sample and reported in S7 Table. In addition, the preprocessing step by BRAQUE allows a precise identification of artifacts (focal tissue loss, scratch, tiling borders, staining focal or partial defects, etc.) as separate clusters with an unique phenotype and spatial distribution, which can be manually excluded from the analysis (S12 Fig.D).

The pipeline discriminate cell types which are closely commixed with others, e.g., CD4 and CD8 T-cells [10], stromal cells and endothelium, sinus-lining macrophages and endothelium etc., without applying computational corrections for signal crossbleeding [144] (S12 Fig.B).

To test the robustness of BRAQUE in the presence of non-specific noise, we run a sample (UPN32) before and after subtracting the AF. As shown in S12 Fig.C, BRAQUE was moderately affected by noise. (S12 Fig.C and S16 Table).

## Antibody panel high dimensional validation

We next tested how a reduced antibody panel would affect the results. Three panels were designed (S6 Table and S12 Fig.E) and run on two whole LN (UPN107 and UPN108):

33 "non-redundant" Abs: lineage-defining Ab with minimal or nil staining across the other lineages (e.g., CD3 and CD20).

41 "TF and friends" Abs: all the anti-TF Ab and a selection (16) of lineage-defining Ab

68 "only clean" Abs: Ab chosen for high signal-to-noise ratio.

Compared to the full Ab panel, each of the reduced panels had variations in detection or representation (S17 Table). The TF panel was the most affected, having the highest junk and uninterpretable results. Some cell types (Treg, plasma cells, RORC) had the most consistent results, probably because of a combination of unique highly characterizing markers and a distinct phenotype. We also found that a minimum of ~68 markers (S12 Fig.E and S6 Table) is required for the cell classification of most hematolymphoid cells.

An analysis of the TMA cores with Seurat [145] produced a 23 cell type classification of the LN analogous to BRAQUE (S13 Fig.).

## Sample size representativeness

A comparison of the cell type composition of whole LN sections (WSI) vs TMA cores shows that these latter are representative of the whole sections except for the smallest populations, which were under- or over-represented because of the pathologist-driven targeted sampling, notwithstanding the 2 mm diameter TMA cores (S14 Fig.).

## Iconographic rendering

Computer-generated .png files were re-colored, modified and assembled with the following softwares:
Fiji (RRID:SCR_002285) (https://imagej.net/ last accessed January 15 2024) [146].
Adobe Photoshop 25.3.1 (RRID:SCR_014199).
Adobe Illustrator 28.1 (RRID:SCR_010279).
RGB composite immunofluorescent images were imported as grayscale images into Fiji, optimized via the Brightness/contrast tool automatic function, cropped and colored as desired.

## Quantification and statistical analysis

**High dimensional analysis with BRAQUE.** Data extracted from a tiled set of images after CyBorgh segmentation were processed via a bioinformatic pipeline, BRAQUE [10].

BRAQUE is a python pipeline for automated cluster enhancing, identification, and characterization. The output consists of multiple clusters, whose numerosity is defined by the size of the smallest cluster (usually not below 0.005% of the cell number or ~20 cells). Each cluster is defined by A) markers ranked by probability or possibility to identify the cluster (i.e., a robust measure of effect size comparing within cluster marker distribution with outside cluster marker distribution), B) a tissue map of the cells belonging to the cluster and C) the expression of a pre-defined set of diagnostic markers for that cluster, compared to the whole population. Each cluster is classified by an expert supervision into cell types [105]. The parameters to apply BRAQUE to the LN dataset were tuned by varying the parameters and scoring a sample. Once the results were satisfactory, BRAQUE was applied to the whole dataset (see Supplementary material @ https://doi.org/10.17632/3ntbp3zdzh.2).

## BRAQUE preprocessing

The usage of Lognormal Shrinkage showed consistent and significant improvement in terms of all the following aspects: better cell groups separation in the lower dimensional embedding, more consistent and compact individuated cell types, smaller number of unclear cells (labeled in the "noise" cluster), highly reduced number of spurious "mixed phenotypes" such as CD4 + CD8 + T cells, CD20 + T-cells etc.[10] (See S3 Table and S15 Table).

## Gaussian mixture components

Given $G_i$ as the initial Gaussian mixture components number, and $G_f$ as the final number of components, selected by the algorithm at convergence.

We observed that a higher number of $G_i$ strongly helps to scatter different cell types apart, reaching a plateau in performances around 20~30.

Moreover, we wanted $G_i$ to be higher than $G_f$ for all markers, in order to check the algorithm had enough starting components and was not limited by input parameters (since $G_i$ acts as an upper limit for $G_f$, and the algorithm can drop useless components but not add useful ones if needed).

Lastly we know that higher $G_i$ means significantly higher computation times, and therefore we considered choosing for it the smallest value for which $G_i > G_f$ for all available markers, resulting in 20 for all lymph nodes except for the largest one, UPN109, for which we used 28.

## BRAQUE nested classification

We decided to apply our pipeline in a hierarchical fashion, first by identifying clusters of either B-cells, CD4 + T-cells, CD8 + T-cells, Dendritic cells, Myeloid cells, or other cell types (BRAQUE$^{global}$).

The non-hematopoietic cell types (such as endothelial cells) were definitively assigned to their cluster, while for each of the other subtypes we performed the pipeline once again, apart from the preprocessing step, on a specific subset of markers (S6 Table) (BRAQUE$^{subclass}$). Unclear, unassigned or noise clusters were discarded from the analysis (e.g., noise cluster cells, unclear cell types with contradictory markers, unseparated T-cells etc.)

This decision was due to how most cell types are interpreted and defined. A very specific and selective set of markers is necessary to sub-specify, e.g., two B-cells or two CD4 T-cells, being the rest of the panel unnecessarily noisy and uninformative in the analysis.

In the BRAQUE$^{subclass}$ analysis, the HDBSCAN "-1' cluster refers to an "average" cell of the cell type analyzed, not to cells of unassigned clusters, as in BRAQUE$^{global}$, which were excluded from BRAQUE$^{subclass}$ analysis.

Five whole LNs were used for the analysis of CD4, CD8 and B cell types, six for the other cells.

## Minimum size of cell group

Minimum size of a cell group is a parameter which needs to be tuned for new hyperplexed images dataset and new antibody panels. In our case, reducing this number might lead to more, clearer, and smaller clusters, while raising it causes more cells to enter the "unclear" cluster. Tuning these parameters based on the number of resulting clusters led to faster and better separation of clearer cell types, and therefore we assessed it in the range of 100~300 cells, which led to 50~100 clusters for each cell subtype among B-cells, CD4 T-cells, CD8 T-cells, Dendritic cells, and Myeloid cells. The smallest cluster (10 cells) consisted of CD8 naïve T-cells, the largest (101,180 cells) of IgD + CD27neg B-cells.

## BRAQUE data output and analysis

Each cluster was evaluated by examining the following types of data:

*i*) the comprehensive list of markers significantly expressed with respect to the average sample distribution (via a Welch t-test),

*ii*) the ranking of said markers according to their decreasing effect size (so to have overexpressed markers as most characteristic ones),

*iii*) the expression of selected lineage-defining markers (these independently from the ranked list) and

*iv*) the spatial location within the tissue.

The output of the pipeline has been described before [10,105].

Since no signal thresholding was performed, intersample pre-analytical variations (fixation, processing etc.) which may modify the tissue antigenicity, would not affect the relative ranking of cluster-defining markers across samples.

Clusters were classified according to the similarity to known phenotypic consensus profiles in published papers. Clusters with incoherent markers were classified as "junk" and discarded. Phenotypes not published before were either classified as "undefined" or grouped in homogeneous new phenotypes. Classified clusters were grouped into individual cell types.

This process was performed independently by two investigators (GC and GEM) and harmonized.

In order to build a comprehensive list of markers for each cell type, two type of data were extracted from the ranked list of markers of each cluster (Table 1, S18 Table and S19 Table): the list of markers present in 70–100%, 50–69% and 30–49% of all clusters belonging to a given cell type and the list of the markers ranked 1st to 5th in that cell type. To compute this latter, we calculated the median value of the position of a given marker in each ranked list for the clusters belonging to that cell type and listed the makers ranking from 1st to 5th place.

The two types of data provide independent information: the first lists the markers identifying a given cell type, independently of the signal-to-noise ratio, the ranking or the statistically significant effect size. The second reflects the effect size and, combined with the first, which marker best applies to flow-cytometry or visual immunostains for cell detection in tissue; it may be used as a proxy for signal brightness, but may represent only a fraction of the clusters of that cell type.

Further analysis of how BRAQUE compares with previous pipelines has been reported [105].

## Post-classification data analysis

**Neighborhood analysis.**  We computed the statistical significance of the neighborhoods with two different approaches: first with a fisher test and then with a newly devised method based on cellular "overlap".

The reason behind this double approach resides in the fact that the former represents a classical/dichotomic approach (near/far, significant variation in odds ratios), while the second highlights a rather continuous measure (how near? More than randomly expected?), giving different importance to neighboring cells based on their spatial closeness (Fig 14).

The overlap test briefly consists of assigning each cell an interaction radius, and then compute the observed and expected amount of pixels (i.e., infinitesimal elements of area) covered by both interaction radii of neighboring cells for different populations (or even same for same population overlap). The pixel-hitting procedure is the equivalent of a Bernoullian process, with success probability given by the ratio between cell interaction area and whole lymph node area. Therefore the overlap is computed using Binomial cumulative density functions.

To increase statistical robustness of outcomes, the analysis was repeated using 2 different distance thresholds to define neighbors, specifically 10μm and 20μm [147].

In other words, neighboring effects were considered significant only if the analysis using 10μm (22 pixels) threshold and the

analysis using 20μm (44 pixels) threshold were both significant for the given neighboring effect over all available samples independently. Both tests were corrected for multiple tests with Bonferroni correction.

Considering cell population A and B, this translates into taking the maximum of the p-values measured across all samples with both populations, and consider it significant only if the maximum p-value was below the Bonferroni corrected significance threshold.

These measures (max p value, together with Bonferroni correction, considering 6 lymph nodes each with 2 different thresholds) allowed for maximal protection against false positives, meaning that all the reported outcomes had been consistent over all samples and with highly significant performances.

Together with statistical tests, we wanted to evaluate a further metric, the median distance to the nearest cell of and from each specific population (Fig 14, Figs S8-S10 Figs.), in order to provide further information.

## Overlap

We start by defining an interaction radius for each cell, by drawing this imaginary interaction circle, centered on every cell, so we can quantify spatial neighboring through increasing shared area between interaction circles of neighboring cells, where the closer the cells, the higher the overlapping area. Therefore if we use $r = 22$ pixels (10 μm at 0.45 μm/pixel) as interaction radius, two cells will interact if they are closer than 1 diameter (44 pixels or 20 μm), since both cells have a circle of $\pi r^2$ around them), with a possible overlap going theoretically from 0 up to 100% (where upper limit is never reachable due to cell impossibility to compenetrate each other).

Lastly, we can build our own statistical overlapping test, by considering the chance of a pixel (i.e., an infinitesimal element of area!) being hit by an interaction circle of a cell as: a bernoulli event with success probability $\mathbf{p}$ given by: $\mathbf{p} = \mathbf{a/A}$, where $\mathbf{a}$ is the area of the interaction circle ($\pi r^2$ with $r = 44$ pixels or 22 pixels, depending on the used threshold) and $\mathbf{A}$ is the total area of the available tissue (computed by measuring total interacting area covered by cells from the same sample).

Since $\mathbf{a} \ll \mathbf{A}$ we can ignore border effects on $\mathbf{A}$ and cell sizes, and now for each cell type pair $\mathbf{(i,j)}$ we can do the following:

Iff $\mathbf{j} = \mathbf{i}$:

- draw an interaction circle of area $\mathbf{a}$ for every cell of type $\mathbf{i}$;

- measure $\mathbf{o}_i$ as the how many pixels are hit by at least 2 interaction circles;

- compare $\mathbf{o}_i$ with its asymptotic distribution under the null hypothesis, i.e. $\mathbf{o}^{TH}_i = binomial(\mathbf{n} = \mathbf{n}_i, \mathbf{p} = \mathbf{a/A}, \mathbf{k} > 1) / binomial(\mathbf{n} = \mathbf{n}_i, \mathbf{p} = \mathbf{a/A}, \mathbf{k} > 0)$, measurable using bootstrap methods, to estimate how likely would be for random cells to overlap as much as the observed ones for population $\mathbf{i}$.

Iff $\mathbf{j} \neq \mathbf{i}$:

- draw an interaction circle for every cell of type $\mathbf{i}$, and for type $\mathbf{j}$;

- define population areas $\mathbf{a}_i$ and $\mathbf{a}_j$ as all pixels hit by at least 1 cell of population $\mathbf{i}$ (or $\mathbf{j}$);

- measure $\mathbf{o}_{ij} = |\mathbf{a}_i \cap \mathbf{a}_j| / |\mathbf{a}_i \cup \mathbf{a}_j|$;

- compare with $binomial(\mathbf{n} = min(\mathbf{n}_i, \mathbf{n}_j), \mathbf{p} = max(\mathbf{a}_i, \mathbf{a}_j)/\mathbf{A}, \mathbf{k} > \mathbf{o}_{ij} min(\mathbf{n}_i, \mathbf{n}_j))$.

By doing so, it is possible to estimate which cell types significantly overlap their interaction areas, and therefore are consistently neighboring with higher intensity than random.

## Fisher significance test

Without elaborating in detail about a well-known test, we wish to highlight the difference between the Fisher test and the overlap method.

In the Fisher statistical test, we simply split cells in near cells and far cells, based on the selected distance threshold. Therefore there is no different contribution between two cells with different distances, as long as they are both within the threshold, or both outside the same.

What we are obtaining with the Fisher test is the odds ratio, or rather the ratio among the proportion inside and the proportion outside the neighborhood. So for a cell population which has 20% of its neighboring cells as B cells, while outside its neighborhoods there is a proportion of 5% B cells, it will result in an odds ratio of 20% / 5% = 4.

Since this measure can grow from 1 to infinity due to significant enrichments, but only span from 1 to 0 due to significant absences, it is used to symmetrize this by taking the logarithm of the odds ratios, resulting in a measure that goes from negative infinity up to positive infinity, with 0 as neutral (nor enriched nor absent population).

 

## Distance

We further investigated cell interactions by taking another quantitative measure: the median distance from the nearest cell of a given type.

This measure has not an associated statistical test, therefore its only purpose was to quantify how far is usually the nearest cell of type B when we start from a cell of type A, and this is what is reported in S10 Fig.

## Significance of zonal distribution of cell subsets across zones

The cell number and the zonal location for each cell type were computed. The "interstitial" type of distribution, not specific for a single LN zone, was not informative and therefore not included in the analysis from the regional point of view, but its cells were used for all enrichment tests as part of the "outside region" cells.

For each pair of region and cell type, a fisher test was conducted and results were aggregated as previously done for neighborhood tests: average odds ratios among different samples, consider as significant only those having consistently the same sign (always enriched or always depleted) with a max p value across samples lower than bonferroni corrected significance threshold. This allowed for the maximum possible control of false positives, stating as significant only cell types which were consistently and heavily enriched/depleted over the same region. Sample averaged significant odds ratios can be observed in Fig 15.

In order to further explore all available information, 2 more simple quantitative analyses were conducted: cell type relative distribution over available region, and average regional composition in terms of present cell types. Both can be observed in Fig 15. These analyses, even if not statistically validated, offer a view over cell types predisposition to occupy or not certain regions and viceversa, and combined with the enrichment test they provide a more complete information.

## Supporting information

**S1 File. Supplementary figures.**
(PDF)

**S1 Table. Clinicopathologic data of the patients and samples.**
(XLSX)

**S2 Table. Primary and secondary antibody table.**
(XLSX)

**S3 Table. Cell classification of whole LN and TMA cores with BRAQUEglobal (BRAQUE1).**
(XLSX)

**S4 Table. Criteria to group cell types classified by BRAQUE1 for BRAQUE2 analysis.**
(XLSX)

**S5 Table. Comprehensive granular classification of LN cells (BRAQUE1).**
(XLSX)

**S6 Table. Antibody list and subpanel allocations.**
(XLSX)

**S7 Table. Cluster classification: all clusters for whole LNs.**
(XLSX)

**S8 Table. Comprehensive catalog of cell types.**
(XLSX)

**S9 Table. Neighborhood relationship between cell types.**
(XLSX)

**S10 Table. Overlap between cell types.**
(XLSX)

**S11 Table. List of cases in which TCF7+PAX5+cells are found.**
(XLSX)

**S12 Table. Spatial frequency allocation of cell types.**
(XLSX)

**S13 Table. Presence of Fairy Circles in whole LN sections and 2 mm TMA cores, with clinicopathologic data.**
(XLSX)

**S14 Table. Definitions and description of LN anatomical zones.**
(XLSX)

**S15 Table. Comparison of cell segmentation with CyBorgh or CellPose2.**
(XLSX)

**S16 Table. Comparison of cell clustering by BRAQUE on a TMA core with noise.**
(XLSX)

**S17 Table. Effect of Ab panel variations on the cell classification on two whole LN, UPN107 and UPN108.**
(XLSX)

**S18 Table. Comprehensive granular classification of LN cells.**
(XLSX)

**S19 Table. Harmonized subsets by broad cell types.**
(XLSX)

## Acknowledgments

We wish to thank Ulf Klein (School of Medicine, University of Leeds, UK), William Vermi and Silvia Lonardi (UNIBS, Brescia, Italy) for a critical review and suggestions, Cecilia Dominguez-Conde and Altea Gjurgjaj (Human Technopole, Milan, Italy) for help with data integration, Bachir Alobeid (Columbia U., USA), Carlo Parravicini (Ospedale L. Sacco, Milan, Italy), Asier Antoranz-Martinez, Lukas Marcelis, Thomas Tousseyn and Johanna Vets (KUL, Leuven, Belgium) for help in the early phase of this research.

## Author contributions

**Conceptualization:** Maddalena M Bolognesi, Lorenzo Dall'Olio, Giorgio Cattoretti.

**Data curation:** Maddalena M Bolognesi, Luisa Lorenzi, Mario Faretta, Giorgio Cattoretti.

**Formal analysis:** Maddalena M Bolognesi, Lorenzo Dall'Olio, Giulio Eugenio Mandelli, Simone Borghesi, Giorgio Cattoretti.

**Funding acquisition:** Francesca M Bosisio, Gastone Castellani, Giorgio Cattoretti.

**Investigation:** Maddalena M Bolognesi, Lorenzo Dall'Olio, Luisa Lorenzi, Ann M Haberman, Mario Faretta, Giorgio Cattoretti.

**Methodology:** Maddalena M Bolognesi, Lorenzo Dall'Olio, Simone Borghesi, Mario Faretta, Gastone Castellani, Giorgio Cattoretti.

**Project administration:** Giorgio Cattoretti.

**Resources:** Giulio Eugenio Mandelli, Luisa Lorenzi, Francesca M Bosisio, Ann M Haberman, Govind Bhagat, Giorgio Cattoretti.

**Software:** Maddalena M Bolognesi, Lorenzo Dall'Olio, Simone Borghesi, Mario Faretta.

**Supervision:** Gastone Castellani, Giorgio Cattoretti.

**Validation:** Giulio Eugenio Mandelli, Mario Faretta.

**Visualization:** Maddalena M Bolognesi, Lorenzo Dall'Olio.

**Writing – original draft:** Maddalena M Bolognesi, Lorenzo Dall'Olio, Giulio Eugenio Mandelli, Francesca M Bosisio, Ann M Haberman, Govind Bhagat, Simone Borghesi, Giorgio Cattoretti.

**Writing – review & editing:** Giorgio Cattoretti.

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
