## [Decision Letter · Decision Letter 0]

2 Feb 2026

Dear Dr. Cattoretti,

Thank you for submitting your manuscript to PLOS ONE. After careful consideration, we feel that it has merit but does not fully meet PLOS ONE’s publication criteria as it currently stands. Therefore, we invite you to submit a revised version of the manuscript that addresses the points raised during the review process.

plosone@plos.org. . . . A letter that responds to each point raised by the academic editor and reviewer(s). You should upload this letter as a separate file labeled 'Response to Reviewers'.A marked-up copy of your manuscript that highlights changes made to the original version. You should upload this as a separate file labeled 'Revised Manuscript with Track Changes'.An unmarked version of your revised paper without tracked changes. You should upload this as a separate file labeled 'Manuscript'.

We look forward to receiving your revised manuscript.

Kind regards,

Deborah S. Cunninghame Graham

Academic Editor

PLOS One

Journal Requirements:

“Regione Lombardia POR FESR 2014–2020, Call HUB Ricerca ed Innovazione: ImmunHUB. (GCat and MMB)

EU Horizon 2020 programme (GenoMed4All project #101017549, HARMONY and HARMONY-PLUS project #116026) (GCast)

The AIRC Foundation (Associazione Italiana per la Ricerca contro il Cancro; Milan, Italy; projects #26216. (GCast)

KUL INTERNE FONDSEN MIDDEL-Zware infrastructuren EMH-D8191-AKUL/19/30 I005920N (FMB)

FWO Fundamenteel Klinisch Mandaat EMH-D8972-FKM/20. (FMB)

The National Recovery and Resilience Plan (NRRP), Mission 4 Component 2 Investment 1.4 - Call for tender No. 3138 of 16 December 2021, rectified by Decree n.3175 of 18 December 2021 of Italian Ministry of University and Research funded by the European Union – NextGenerationEU; Project code CN_00000033, Concession Decree No. 1034 of 17 June 2022 adopted by the Italian Ministry of University and Research, CUP B83C2200293000 , Project title “National Biodiversity Future Center - NBFC”. (MMB)”

4. Please be informed that funding information should not appear in the Acknowledgments section or other areas of your manuscript. We will only publish funding information present in the Funding Statement section of the online submission form. Please remove any funding-related text from the manuscript.

Additional Editor Comments (if provided):

The manuscript needs minor revisions - as laid out in the comments by reviewer 2. Please can you ensure that you clarify which findings the authors consider the most biologically significant, and how these results relate to prior knowledge or to findings from alternative approaches such as scRNA-seq or spatial transcriptomics.

Major comments:

Given the large number of cell types defined, it would be helpful to clarify whether each represents a biologically distinct population or whether some reflect minor variations around an individual baseline phenotype.

For each cell type, could the authors quantify inter-individual variability in spatial localisation across lymph nodes? In particular, Fig. 11 suggests substantial differences in the localisation of distinct dendritic cell subsets between samples. However, it is unclear to what extent this reflects true biological variability versus differences arising from the plane of tissue sectioning.

For cell types not previously described, or not previously described in situ, could the authors further validate these populations? For example, this could include staining adjacent sections from the same lymph nodes with additional markers, examining lymph nodes from additional donors, or identifying analogous populations in previously published datasets.

Throughout the manuscript, the authors could more clearly relate their findings to previous studies of human lymph nodes and clarify how the cell types they define correspond to those described in other work. Several groups have mapped lymph node cell populations using approaches such as scRNA-seq and spatial transcriptomics, each with distinct strengths and limitations. A more explicit comparison would help readers understand which aspects of lymph node biology are best resolved by multiplexed immunofluorescence versus transcriptomic methods, and where the results are concordant or divergent.

How did the authors decide which clusters to group together into a single cell type? How much phenotypic variability exists among the clusters grouped into each cell type, and how can one be confident that these represent a single biological population rather than a mixture of related but distinct cell types?

How much inter-individual variability was observed in cellular neighbourhoods, and is this variability accounted for statistically? Which neighbourhood relationships were highly consistent across donors, and which appeared donor-specific?

Minor comments:

Not all figures are referenced in order in the text.

HDBSCAN appears to identify a large number of very small clusters (e.g. for myeloid cells in Fig. 2D). How should these small clusters be interpreted biologically? By how many markers do these clusters typically differ, and how sensitive are the results to the choice of clustering algorithm or parameters?

Did you identify any association between, for example age, and cell type frequency?

Table 2 and 3 – how were the markers chosen for CD4 and CD8 classification (e.g. TCF7, FOXP3, TIM3, PD1, CD137, Ki67)? Clarifying this rationale would improve reproducibility and interpretability.

The cluster labels in Fig. 3 align with those in Fig. 4. However, the strategy for defining these cell types using TCF7 in combination with FOXP3, checkpoint molecules, and Ki67 is not described in the text until later (when introducing Fig. 4) – this is confusing for the reader.

Line 269-270 – the authors describe effector cells as progenitor exhausted (Tpex). Given that this is a healthy human LN cohort, would one expect Tpex cells to be present at such frequencies? If so, additional discussion or justification would be helpful.

Line 373-374 – given that some CD5⁺ subsets lack canonical B cell markers, how can their B cell identity be confirmed (for cells not expressing IgD)? Is it possible that some of these clusters represent mixed populations?

Line 515 – given that ILC3s are reported in situ, could the authors validate the localisation of this population by staining adjacent sections with additional ILC3 markers, or by comparison with published spatial datasets?

Reviewers' comments:

Reviewer's Responses to Questions

**Comments to the Author**

1. Is the manuscript technically sound, and do the data support the conclusions?

Reviewer #1: Yes

Reviewer #2: Yes

2. Has the statistical analysis been performed appropriately and rigorously?

Reviewer #1: Yes

Reviewer #2: Yes

3. Have the authors made all data underlying the findings in their manuscript fully available?

Reviewer #1: Yes

Reviewer #2: Yes

4. Is the manuscript presented in an intelligible fashion and written in standard English?

Reviewer #1: Yes

Reviewer #2: Yes

Reviewer #1: This paper serves as a comprehensive resource. The authors compiled a geographical atlas of the normal human lymph node by staining sections from 19 lymph nodes with a 78-marker antibody panel and analyzing the data using the BRAQUE pipeline. The lymph nodes were considered normal based on their small size and the absence of pathological features.

Both the MILAN staining method—which involves multiple cycles of staining and stripping—and the BRAQUE bioinformatic pipeline were validated. The antibody panel included markers of cell differentiation, chemokine receptors, and transcription factors.

In total, 77 distinct cell types were identified, including T cells, B cells, dendritic cells, innate immune cells, and stromal cells. Their spatial localization within the major, well-defined lymph node compartments was described.

Single-cell RNA sequencing technologies have shown that cells with similar phenotypes can exhibit different functional states, leading to the identification of novel cellular populations. This atlas provides a framework to investigate these populations in situ, and their relative distribution within lymph nodes may help elucidate their functional roles in vivo, within the tissue where immune responses occur.

The paper is complex, but the extensive figures and tables presented in both the main text and supplementary materials are essential for data completeness and for facilitating reuse by other researchers.

Reviewer #2: Summary:

The authors use a high-dimensional cyclic immunofluorescence approach, specifically MILAN, to study cell populations and their spatial organisation in healthy human lymph nodes. Overall the paper is comprehensive and provides a rich and useful resource. However, it is not immediately clear which findings the authors consider the most biologically significant, nor how the results relate to prior knowledge or to findings from alternative approaches such as scRNA-seq or spatial transcriptomics.

Major comments:

Given the large number of cell types defined, it would be helpful to clarify whether each represents a biologically distinct population or whether some reflect minor variations around an individual baseline phenotype.

For each cell type, could the authors quantify inter-individual variability in spatial localisation across lymph nodes? In particular, Fig. 11 suggests substantial differences in the localisation of distinct dendritic cell subsets between samples. However, it is unclear to what extent this reflects true biological variability versus differences arising from the plane of tissue sectioning.

For cell types not previously described, or not previously described in situ, could the authors further validate these populations? For example, this could include staining adjacent sections from the same lymph nodes with additional markers, examining lymph nodes from additional donors, or identifying analogous populations in previously published datasets.

Throughout the manuscript, the authors could more clearly relate their findings to previous studies of human lymph nodes and clarify how the cell types they define correspond to those described in other work. Several groups have mapped lymph node cell populations using approaches such as scRNA-seq and spatial transcriptomics, each with distinct strengths and limitations. A more explicit comparison would help readers understand which aspects of lymph node biology are best resolved by multiplexed immunofluorescence versus transcriptomic methods, and where the results are concordant or divergent.

How did the authors decide which clusters to group together into a single cell type? How much phenotypic variability exists among the clusters grouped into each cell type, and how can one be confident that these represent a single biological population rather than a mixture of related but distinct cell types?

How much inter-individual variability was observed in cellular neighbourhoods, and is this variability accounted for statistically? Which neighbourhood relationships were highly consistent across donors, and which appeared donor-specific?

Minor comments:

Not all figures are referenced in order in the text.

HDBSCAN appears to identify a large number of very small clusters (e.g. for myeloid cells in Fig. 2D). How should these small clusters be interpreted biologically? By how many markers do these clusters typically differ, and how sensitive are the results to the choice of clustering algorithm or parameters?

Did you identify any association between, for example age, and cell type frequency?

Table 2 and 3 – how were the markers chosen for CD4 and CD8 classification (e.g. TCF7, FOXP3, TIM3, PD1, CD137, Ki67)? Clarifying this rationale would improve reproducibility and interpretability.

The cluster labels in Fig. 3 align with those in Fig. 4. However, the strategy for defining these cell types using TCF7 in combination with FOXP3, checkpoint molecules, and Ki67 is not described in the text until later (when introducing Fig. 4) – this is confusing for the reader.

Line 269-270 – the authors describe effector cells as progenitor exhausted (Tpex). Given that this is a healthy human LN cohort, would one expect Tpex cells to be present at such frequencies? If so, additional discussion or justification would be helpful.

Line 373-374 – given that some CD5⁺ subsets lack canonical B cell markers, how can their B cell identity be confirmed (for cells not expressing IgD)? Is it possible that some of these clusters represent mixed populations?

Line 515 – given that ILC3s are reported in situ, could the authors validate the localisation of this population by staining adjacent sections with additional ILC3 markers, or by comparison with published spatial datasets?

.

Reviewer #1: **Yes:** Rita CarsettiRita CarsettiRita CarsettiRita Carsetti

Reviewer #2: No

---

## [Author Response · Author response to Decision Letter 1]

12 Feb 2026

Additional Editor Comments (if provided):

The manuscript needs minor revisions - as laid out in the comments by reviewer 2. Please can you ensure that you clarify which findings the authors consider the most biologically significant, and how these results relate to prior knowledge or to findings from alternative approaches such as scRNA-seq or spatial transcriptomics.

*- The Reviewers and the Editor made significant comments and observations, which we truly appreciated.

Major comments:

Given the large number of cell types defined, it would be helpful to clarify whether each represents a biologically distinct population or whether some reflect minor variations around an individual baseline phenotype.

*- Our study is phenotypic and not functional, as stated in lines 143-144 (initial submission): “... We here provide a granular purely phenotypic classification…..”. The chosen markers have been historically selected to represent phenotypic traits (CD4, CD8, CD27, etc.) useful for classification of biological entities, however the boundaries between pure phenotypic and functional biological markers (CD25, Ki-67, bcl-2, etc.) are blurred. Hence the need of a more precise nomenclature, which is a general need: see Comet, N. R. & Gerlach, C.[1] : “The T cell subsetting challenge”.

As per the variations, BRAQUE clusters classification is based on the statistical relevance of markers, not on the staining intensity and the variations may be a mix of biology and technicalities.

For each cell type, could the authors quantify inter-individual variability in spatial localization across lymph nodes? In particular, Fig. 11 suggests substantial differences in the localization of distinct dendritic cell subsets between samples. However, it is unclear to what extent this reflects true biological variability versus differences arising from the plane of tissue sectioning.

*- These are indeed very poignant points.

Re: quantify individual variability; quantification of this variability is shown by the third column in Tables 2-6 (“Location (% of cells)”) and Table S17, S18, S20, where the percentage of cells located in discrete anatomical regions is listed.

Re: Biological variability vs the plane of tissue sectioning; all the LN sections we examined are leftovers, part of a clinical diagnostic procedure where the small specimens are sectioned along planar major axis, in order for the section to be representative. They are also small and judged devoid of pathological features by the pathologist in charge of the clinical diagnosis. Despite this, there is obvious heterogeneity which we interpreted as true biological variability, rather than sampling errors. We tried to correct for this variability by:

- Analyzing the whole section

- Considering only cell types or structures (e.g. Fairy Circles) present in at least two LN

- Considering neighborhood and cell overlap values only if statistically significant in all LN examined and measured on both 10µm and 20µm distance radius (see Figs S8 and S9).

Thus, our results may be a conservative representation of the biological variability existing in LN considered “normal”. The localization of each individual cluster in spatially-defined LN structures (see Table S15), allows to estimate this variability.

2D sections have limitations, as Reviewer #2 points out. 3D multiplex staining and analysis[2] is the ideal technique, however, as per the wording of Yapp et al. “The 3D tissue imaging is unlikely to be a replacement for 2D approaches, particularly in translational research involving large sets of specimens, because 3D imaging is harder to perform, generates larger datasets and is not directly compatible with existing histopathology work-flows. Instead, 3D data are likely to be most useful for detailed and precise study of a more limited number of samples.”

For cell types not previously described, or not previously described in situ, could the authors further validate these populations? For example, this could include staining adjacent sections from the same lymph nodes with additional markers, examining lymph nodes from additional donors, or identifying analogous populations in previously published datasets.

*- Overall, five cell types out of 77 (6.4%; CD7/23/GZMB Myelo, GZMB+, Act Macs, PU1 cMAF, perifollicular stroma) have not been described before. Cell types not previously described or located are reported only if present in several LN.

For the remaining cell types, we referenced previously published analogous populations in each table in the last column on the right. We tried to limit the number of quoted references to the essential from a very very large body of knowledge, and we apologize if we missed key quotes.

By which factor would an increase of sample number, either TMAs or whole sections, substantially add to the present data to further validate these populations?

The markers chosen are i) validated for FFPE use, ii) with the most differential expression (with limits; see Fig. S1D), iii) redundant for cell identification. Additional markers, assuming that there are FFPE-proof validated Ab available, will necessarily require to be stained in multiplexing with at least >60 diagnostic antibodies and the value of these additions may be worth for one cell type at a time only. Not to mention the magnitude of such an effort.

Throughout the manuscript, the authors could more clearly relate their findings to previous studies of human lymph nodes and clarify how the cell types they define correspond to those described in other work.

*- See our comment above

Several groups have mapped lymph node cell populations using approaches such as scRNA-seq and spatial transcriptomics, each with distinct strengths and limitations. A more explicit comparison would help readers understand which aspects of lymph node biology are best resolved by multiplexed immunofluorescence versus transcriptomic methods, and where the results are concordant or divergent.

*- We are aware of the growing list of groups who have mapped lymph node cell populations using approaches such as scRNA-seq and spatial transcriptomics.

As soon as we established our LN cell classification, we reached out to the lab of Cecilia Dominguez-Conde, the leading Author of reference Science papers[3, 4]. We attempted to perform mosaic integration[5] of our data with scRNAseq or CITE-seq data from the LN samples in her dataset; we obtained inconclusive results (besides the integration of the broad usual cell types), because of the difficulties in matching the level of depth of the annotation in each datasets and the limited number of cells available for comparison (a known magnitude difference between single cell suspension and in-situ datasets). She is acknowledged in the ms.

We then screened ~170 papers analyzing SLOs (mouse + humans) with single cell technologies, selecting a dozen specifically dealing with human LN and using either one or both of CITE-seq, spatial transcriptomics (CosMx SMI, Visium, Nanostring) and spatial proteomics (IBEX, CODEX)[3, 4, 6-15]. We also reviewed a few significant papers using scRNA-seq on SLOs. [16-20].

What we noticed is:

Most papers (with few exceptions[6-8]) produce a <50 cell type classification, way too coarse for a meaningful comparison with our data.

None of the high-classificators (≥50) match the granularity of our data e.g. on T cells or B cell subsets. The most glaring examples are a Tonsil cell atlas[7] (121 cell types) and a NSCLC LN classification paper[6] (50 cell types), which have CD5 in the ADT CITE-seq panel and fail to identify CD5+ B cell, CD5+ dendritic cell subsets and other subsets.

A paper dedicated to “conventional” scRNA-seq analysis of B-cell development[20] (18 B-cell types) fails to identify the B1 cells we describe and are reported by a paper using CITE-seq[4].

Multiple papers note that proteomic data such as CITE-seq are necessary if not obligatory to classify human cells[6, 9, 14, 21, 22].

That mRNAs and proteins can be used interchangeably for cell classification is dubious at best[10, 21-28].To quote one of these papers[22] “Previous studies have revealed poor mRNA–protein correlation in T cells regardless of functional status and the importance of post-transcriptional regulation in T cell differentiation and function. .. These results underscore the importance of directly defining the proteome rather than inferring it from the transcriptome.”

A recent example of measurement of RNAs and proteins in single neurons[29] states that”... We document extensive transcriptome–proteome discordance across cell types, particularly in genes associated with neurodevelopmental disorders. Proteins exhibit markedly higher cell-type specificity than their mRNA counterparts, underscoring the importance of proteomic-level analysis.”

We believe that the burden of the proof for the ability to classify cells is on transcriptomics and not on proteomic methods, which represent the benchmark on which to validate other assays.

How did the authors decide which clusters to group together into a single cell type? How much phenotypic variability exists among the clusters grouped into each cell type, and how can one be confident that these represent a single biological population rather than a mixture of related but distinct cell types?

*- The rationale for designing BRAQUE is “the idea that merging similar clusters is easier than splitting unclear ones into clear subclusters.”[30]. As described in the M&M section, we proceeded by merging clusters which i) were similar by phenotype (although each characterizing marker would be ranked differently in each cluster), ii) correspond to a phenotype already described in the literature, iii) had a consistent space allocation. This was done manually and over multiple rounds of sub-clustering. The cell type-defining markers and their frequency is reported in three columns in the tables. This process has been described in a recent publication[31] (see ibid Chapter 2.12). Differently from conventional flow-cytometry and imaging techniques, dimensionality reduction algorithms dispose of gating[32]; we used i) pure statistical probability that a marker contributes to define a cluster, ii) its effect factor, a proxy of signal intensity, iii) a plot of the indicized marker distribution curve in the cluster, compared to the whole sample, for a few canonical markers[30]. Negative markers are not used in these algorithms, although BRAQUE provides a ranking of the most NOT-statistically relevant markers.

Note that HDBSCAN/BRAQUE allocate statistically uninterpretable phenotypes in a special group (named -1) which we discarded.

The expression of each marker inside a single cluster may vary not only because of biological variability but also because of technical factors (segmentation, section thickness, etc.). Keep in mind that we i) batch processed the slides[33] for staining and ii) used saturating staining conditions[34] in order to minimize variability due to staining.

We took the expression of literature-published, lineage-defining, redundant markers in each cluster to assign each one to a cell type. The process was done on the whole sample first (BRAQUEglobal, see Table S16) and subsequently on broad cell types (BRAQUEsubclass), as per the manuscript. We took a conservative approach and kept the established phenotypes as a reference as much as possible.

The data showing the variability are reported in Tables 1-7 and Tables S4, S16, S18 and S19. Look for the columns reporting markers frequency in cell type-defined clusters or by cell type-defined numbers (Table S18). The tables also report the number of clusters and the total number of cells per each cell type. The phenotype of individual clusters for BRAQUEsubclass are available in Table S15.

As per the hypothesis of cell type mixtures, read here below.

The explosion in granularity that BRAQUE produces makes us believe that none of the clusters represent a mixed population. This Reviewer however may mean “mixed signals” population, i.e. signal bleeding from nearby cells.

Hyperplexed staining literature data on dense lymphoid tissue show that about a quarter (25.55%) of the cells are spurious doublets because of crossbleeding and this number can be somewhat lowered (18.93%) by using an image preprocessing algorithm (REDSEA[35]). Another example of failure to reduce the number of spurious double-phenotype cells, despite improvement in segmentation and analysis, can be found in Hunter, B. et al.[36]. A recent paper[37] proposes a laborious image pre-processing set of actions, leading to a reduction of such spurious cell identification. This process also significantly reduces the identification of small subpopulations (ibid Fig S3).

Cyborgh segmentation + BRAQUE somehow solve the problem altogether: spurious CD4/CD8 “cells” account for 0.4% ±1.4% (n=14) (Table S3). These data shows that even if the segmentation may not be optimal, the BRAQUE algorithm is able to counterweigh the neighboring cell contribution more efficiently than a pre-processing like REDSEA or ImmuneCite.

It has been published that admixture of stromal and endothelial cells cause the greatest disagreement with ground-truth annotation with sophisticated pipelines such as STELLAR[38]. This does not seem to be our case, given the ability of CyBorgh+BRAQUE to resolve neighboring cells as shown in Fig.S12. See also how BRAQUE dissects stromal cells in a LN database stained and segmented by the source (CODEX; [31]).

In other words, the lognormal shrinkage preprocessing that BRAQUE does, coupled by the statistical-based relevance assigned to each marker’s signal, accomplish what has been proposed for in situ transcriptomics to correct for mis-segmentation[39]. The limitation is obviously due to the technique, as incoming reports on spatial transcriptomics also show[39].

The results we obtained disappointingly fail to identify subtler phenotypic subtypes of established entities, as this Reviewer may suggest, despite producing the largest number of cell types described so far in-situ in a secondary lymphoid organ.

How much inter-individual variability was observed in cellular neighbourhoods, and is this variability accounted for statistically? Which neighbourhood relationships were highly consistent across donors, and which appeared donor-specific?

*- The criteria to define neighborhood across multiple samples are very conservative: data must be i) statistically significant AND ii) significant in all the samples hosting that cell type. These criteria exclude donor-specific data. Note that the neighborhood was measured on both 10µm and 20µm distance radius (see Figs S8 and S9), excluding spurious results.

Minor comments:

Not all figures are referenced in order in the text.

*- We double checked the observation of this Reviewer, but it seems that the correct order of figure quotation is maintained. What may cause confusion is the place of insertion of the main figures in the text, but this is a submitting necessity and not the final layout.

HDBSCAN appears to identify a large number of very small clusters (e.g. for myeloid cells in Fig. 2D). How should these small clusters be interpreted biologically? By how many markers do these clusters typically differ, and how sensitive are the results to the choice of clustering algorithm or parameters?

*- Part of these questions have been answered above. As per the last question, we show data in which we used Seurat for clustering (Fig S13) and produce comparable data to BRAQUEglobal (Fig S13C). Outputs as heatmaps as Seurat (and Histocat; not shown) do rely on the human eye perception for marker allocations, which has been shown already to be subjective[40]. We have also shown[30] that Phenograph clustering produces inferior results. See also our reply above about biological variability.

Did you identify any association between, for example age, and cell type frequency?

*- We do not have a sufficient age range and samples per range to investigate this.

Table 2 and 3 – how were the markers chosen for CD4 and CD8 classification (e.g. TCF7, FOXP3, TIM3, PD1, CD137, Ki67)? Clarifying this rationale would improve reproducibility and interpretability.

*- The rationale for T-cell classification

---

## [Decision Letter · Decision Letter 1]

24 Mar 2026

The normal human lymph node cell classification and landscape defined by high-dimensional spatial proteomic

PONE-D-25-63676R1

Dear Dr. Cattoretti,

We’re pleased to inform you that your manuscript has been judged scientifically suitable for publication and will be formally accepted for publication once it meets all outstanding technical requirements.

Kind regards,

Deborah S. Cunninghame Graham

Academic Editor

PLOS One

Additional Editor Comments (optional):

Thank you for submitting your revision. All the reviewer's comments have been addressed - so I recommend acceptance of the manuscript.

Reviewers' comments:

Reviewer's Responses to Questions

**Comments to the Author**

Reviewer #1: All comments have been addressed

Reviewer #2: All comments have been addressed

2. Is the manuscript technically sound, and do the data support the conclusions?

Reviewer #1: Yes

Reviewer #2: Yes

3. Has the statistical analysis been performed appropriately and rigorously?

Reviewer #1: Yes

Reviewer #2: Yes

4. Have the authors made all data underlying the findings in their manuscript fully available?

Reviewer #1: Yes

Reviewer #2: Yes

5. Is the manuscript presented in an intelligible fashion and written in standard English?

Reviewer #1: Yes

Reviewer #2: Yes

Reviewer #1: This paper provides a valuable resource for the study of the human lymph node. The authors' revisions in response to the reviewers' comments have added useful details and improved the precision of the work.

Reviewer #2: (No Response)

.

Reviewer #1: **Yes:** Rita CarsettiRita CarsettiRita CarsettiRita Carsetti

Reviewer #2: No

---

## [Editor Report · Acceptance letter]

PONE-D-25-63676R1

PLOS One

Dear Dr. Cattoretti,

I'm pleased to inform you that your manuscript has been deemed suitable for publication in PLOS One. Congratulations! Your manuscript is now being handed over to our production team.

Kind regards,

on behalf of

Dr. Deborah S. Cunninghame Graham

Academic Editor

PLOS One